



# Flooding in the Mekong Delta: Impact of dyke systems on downstream hydrodynamics

Vo Quoc Thanh[1,2,3], Dano Roelvink[1,2,4], Mick van der Wegen[1,4], Johan Reyns[1,4], Herman Kernkamp[4], Giap Van Vinh[5], Vo Thi Phuong Linh[3]

[1]Department of Water Science and Engineering, IHE Delft, the Netherlands
[2]Faculty of Civil Engineering and Geosciences, Delft University of Technology, the Netherlands
[3]College of Environment and Natural Resources, Can Tho University, Vietnam
[4]Deltares, Delft, the Netherlands
[5]Cuu Long River Hydrological Center, Southern Regional Hydro-Meteorological Center, Vietnam

*Correspondence to:* Vo Quoc Thanh (t.vo@un-ihe.org)

**Abstract**. Building high dykes is a common measure to cope with floods and plays an important role in agricultural management in the Vietnamese Mekong Delta. However, the construction of high dykes cause considerable changes in hydrodynamics of the Mekong River. Therefore, this paper aims to assess the impacts of the high dyke system on water level fluctuation and tidal propagation on the Mekong River branches using a modelling approach. In order to consider interaction

between rivers and seas, an unstructured modelling grid was generated, with 1D-2D coupling, covering the Mekong Delta and extending to the East (South China Sea) and West (Gulf of Thailand) seas. The model was manually calibrated for the flood season of the year 2000. To assess the role of floodplains, scenarios consisting of high dykes built in different regions of the Long Xuyen Quadrangle (LXQ), Plains of Reeds (PoR) and TransBassac were carried out. Results show that the percentage of river outflow at Dinh An sharply increases in the dry season in comparison to the flood season while the other Mekong

estuarine outflows rise slightly. In contrast, the lateral river flows of the Mekong River system to the seas by the Soai Rap mouth and the LXQ decrease somewhat in the dry season compared to the flood season due to overflow reduction at the Cambodia-Vietnam border. Additionally, the high dykes in the regions that are directly connected to a branch of the Mekong River, not only have an influence on the hydrodynamics in their own branch, but also on other branches because of the connecting channel of Vam Nao. Moreover, the high dykes built in the PoR, LXQ and TransBassac regions are the most

important factor for changing water levels at Tan Chau, Chau Doc and Can Tho, respectively. The LXQ high dykes result in an increase of daily mean water levels and a decrease of tidal amplitudes on the Song Tien (downstream of the connecting channel of Vam Nao). A similar interaction is also found for the the PoR high dykes and the Song Hau.

## 1 Introduction

Rivers are the major source of fresh water supply for human use (Syvitski and Kettner, 2011). In addition the fresh water

supply is an important resource for ecosystems. When river discharge exceeds its' bank full discharge, its floodplains inundate. The fluvial floods bring both advantages and disadvantages to local residents. Floods are the main source of fresh water supply





and deliver sediments as a natural and valuable fertilizer source for agricultural crops (Chapman and Darby, 2016). This is an important process in the Mekong Delta since the majority of local citizens is a farmer. In contrast, extreme floods may damage crops and infrastructure.

In order to maintain agricultural cultivation during flood seasons, dyke rings have been built to protect agricultural crops in
the Vietnamese Mekong Delta (VMD). As a result, the river system in the VMD has significantly changed, especially after the severe floods in 2000 (Biggs et al., 2009; Renaud and Kuenzer, 2012). A dense canal system has been created in flood-prone areas to efficiently drain flood waters from the Long Xuyen Quadrangle and the Plains of Reed to the West Sea (Gulf of Thailand) and to the Vamco Rivers, respectively (Figure 1).

Recently large hydraulic structures have been built not only in the flood-prone areas but also in the coastal areas to protect
cropping systems from saline intrusion. Therefore, hydrodynamic processes have considerably changed. Understanding the prevailing hydrodynamics is essential for sustainable water management in these regions.

The high dyke system is intended to reduce local natural flood hazards, but may alter the hazard downstream (Triet et al., 2017). Besides, they also increase potential risk due to dyke breaks. Following different approaches, Tran et al. (2017) found that the high dyke system in the upstream of VMD causes an increase of the peak water levels in the downstream areas.
However, water levels at these downstream stations are highly dominated by tidal motion. In fact, tides may result in an increase of water levels in the central VMD. They evaluated that the high dyke system may be an important factor, but that sea level rise in combination with land subsidence enhances peak water levels at the central stations to a larger extent (Triet et al., 2017). Thus an analysis of tidal fluctuation is needed to investigate water level changes on the Mekong River.

There is a number of large-scale numerical models used for simulating the annual floods, suspended sediment transport and
evaluating impact of dyke construction in the Mekong Delta (Manh et al., 2014; Tran et al., 2017; Triet et al., 2017; Van et al., 2012; Wassmann et al., 2004). For instance, Tran et al. (2017) investigated the impacts of the upstream high dyke system on downstream part of the VMD. By using a MIKE hydrodynamic model for Mekong Delta, they found that the high dykes system in the LXQ can reduce the discharge of the Song Tien, diverting around 7% of the total volume to Song Hau. In addition, the yearly discharge variations have slight effects on the peak of water levels at Can Tho station, while Triet et al.
(2017) found that the high dyke system caused an increase of the flood peaks from 9 to 13 cm at the central VMD stations. Moreover, the development of the dyke system upstream of the VMD reduces flood retention in this area, leading to a rise of 13.5 cm and 8.1 cm in the peak water levels in the downstream part of the VMD at Can Tho and My Thuan, respectively.

The studies mentioned previously evaluated the impact of the high dykes in the LXQ and the high dykes developed until 2011. The impact of the other floodplain regions needs to be considered, including the LXQ, PoR and TransBassac. Additionally,
(Manh et al., 2014; Tran et al., 2017; Triet et al., 2017) used a common version one-dimension model of MIKE11 for the Mekong Delta and the downstream boundaries are defined at the Mekong's river mouths. However, Kuang et al. (2017) found that river flows can contribute to a rise of water level at the river mouths. Thus, in the present study another modelling approach is used in order to address these issues.



This study aims at assessing the impacts of the high dyke system on water level fluctuation and tidal propagation on the Mekong River branches. An unstructured, combined 1D-2D grid, covering the Mekong Delta and extending to the East (South China Sea) and West seas, is used to simulate the flood dynamics in 2000. Simulated scenarios present the impact of high dykes in different regions and the entire VMD. These outcomes will benefit sustainable water management and planning in the VMD.

**Figure 1. Location of the Mekong Delta.**

### 1.1 The Mekong Delta

The Mekong is one the largest rivers in the world (MRC, 2010). It starts in Tibet, China, flowing through five riparian countries and reaches the ocean via originally nine branches but now seven estuaries. It has a length of 4800 km and a total draining catchment area of 795,000 km$^2$ (MRC, 2005). The Mekong Delta starts from Phnom Penh (Figure 1), where the Mekong river is separated into two branches, namely Mekong and Bassac (Gupta and Liew, 2007; Renaud et al., 2013). The Mekong Delta is formed by sediment deposition from the Mekong River, which provides a yearly amount of 416 km$^3$ of water and 73 Mt/year of sediment at Kratie, mainly distributed in the wet season (Koehnken, 2014; MRC, 2005). The Mekong Delta has a complex river network, especially in the Vietnamese part. The Mekong Delta's river network is illustrated in Figure 4. It has resulted from extensive man-made canal development from 1819 onwards (Hung, 2011).

Regarding land resources, the VMD area is about 4 million ha in which three-quarters is used for agricultural production (Kakonen, 2008). The livelihoods of the local citizens are mainly based on agriculture and aquaculture. The river infrastructure has been developing for the priority of agriculture. It provides just over one half of rice yields in Vietnam and provides up to approximately 90% of exported rice yields from Vietnam (GSOVN, 2010). However, the rice cultivation is highly influenced by annual floods (MRC, 2009a).

The most intensive agricultural production in the VMD is found in An Giang province (Figure 4). Although it is also a flood-prone area, the inundation periods are slightly shorter due to flood withdraw to the West Sea. In the deep flooded zones (Long Xuyen Quadrangle and Plain of Reeds), the high dykes were densely built downstream of these zones. A reason for this is that the downstream LXQ and PoR areas experience low flood peaks so the dyke rings do not need to be heightened as the upstream LXQ and PoR.

The Mekong Delta is dominated by a tropical monsoonal climate. There are two dominant monsoons. The southwest monsoon is from May to October, coinciding with the wet season. The other, drier monsoon period is from November to March, followed by a transition period (MRC, 2010). The mean temperature is approximately 26.5º C. Although the climate is seasonally changing, monthly averaged temperature differences are 4ºC between the hottest and coldest months (Le Sam, 1996). However, seasonal rainfalls are drastically different in terms of time and space. Wet seasons contribute approximately 90% of the total annual rainfall intensity, whereas the dry season (from December to April) account for 10% of the total rainfall. The yearly





mean rainfall is about 1600 mm in the VMD. The highest rainfall is found in the western coastal area of the Mekong Delta, ranging between 2000 and 2400 mm. The eastern coast receives about 1600 mm of rainfall, while the lowest rainfall is in the centre of the VMD (Le Sam, 1996).

## 1.2 High dyke development in the Vietnamese Mekong Delta

The Mekong Delta has been modified extensively over the last two decades after the hugely damaging flood of 2000. Noticeable change is the hydraulic infrastructure, especially the dyke development. Before the dykes were built, a dense canal network was developed to drain floods to the West Sea and to clean acid sulphate soils.

Depending on the dyke function, the dykes can be classified into two categories. The low dykes are built to protect the rice harvest of Summer-Autumn crops in August. This is the rising phase of the annual floods. The low dykes allow flood overflows

and inundation of floodplains so the crests of the low dykes are designed to just equal the maximum water level in August. The high dykes are constructed in order to completely prevent the annual floods and enable intensive agricultural production. Generally the high dykes are designed at a crest level of 0.5 meter above the year 2000 flood peak. In An Giang, there are two kinds of high dykes, consisting of usual high dykes and 2-layer high dykes. The former kind of high dykes has a single dyke ring only. The water dynamics just outside of the dyke ring are dominated by floods. These dykes have a straightforward flood

protection function but a high risk of breaching. The latter type highly protects the field inside of the high dykes. They consist of an overall dyke ring and their sub-dykes. Water levels in the high dyke rings are controlled by sluice gates.

Several studies have mapped the high dykes in the VMD by using remotely sensed images (e.g. Duong et al., 2016; Fujihara et al., 2016; Kuenzer et al., 2013). By this method, the high dykes are identified via flooded and non-flooded areas. However, these results are easily affected by water management of the high dyke rings. For example, in An Giang, the high dyke areas

are managed according to the rule of the 3+3+2 cropping cycle. In other words, these areas are cultivated for eight (3+3+2) agricultural crops in 3 years and allowed to inundate during part of the year once every 3 years. Thus the results need to be verified with observations for reliability of the maps.

High dykes were hardly constructed in the VMD before 2000 (Duong et al., 2016). The year 2000 historical flood, particularly, caused enormous damage to infrastructure and residents' properties. After the flood event, the local authorities planned and

built a cascade of high dykes in order to protect the residents and cultivations which are the major livelihood in this region. In addition, the VMD has great potential for intensification of agricultural production. Until 2009 the area of high dykes was about 1,222 km$^2$, covering around 35% of the An Giang province area and this percentage increased to over 40% (about 1,431 km$^2$) in 2011. Dong Thap has a much lower coverage of about 30%, corresponding to an area of 990 km$^2$. Dong Thap has deep flood-prone areas and its soil contains high concentration of sulphates, causing low potential for agriculture (Kakonen, 2008).

Figure 2 and Figure 3 present numbers and areas of high dykes in An Giang and Dong Thap provinces until 2011. In 2011, An Giang and Dong Thap numbers of high dykes compartments were 329 and 657, respectively. The total high dyke area in An Giang was larger than in Dong Thap (about 14 compared to 10 km$^2$ respectively). As a result, the mean area of a high dyke in





An Giang is larger than in Dong Thap. In fact, high dykes are located mainly along the banks of the Song Tien and Song Hau
(Figure 4) where the soils are alluvial (Nguyen et al., 2015).

**Figure 2. Numbers of high dykes and its areas by districts in Dong Thap province.**

**Figure 3. Numbers of high dykes and its areas by districts in An Giang province.**

**Figure 4. Spatial distribution of high dykes which were built until 2011.**

### 1.3 Flood dynamics in the Mekong Delta

The Mekong Delta is spatially separated into inner and outer parts. The former is dominated by fluvial processes, while the
latter is dominated by marine processes, including tides and waves (Ta et al., 2002). The inner delta is low-lying and flat
(Gupta, 2011). The Mekong River supplies approximately 416 km$^3$ of water volume annually, or on average 13,200 m$^3$/s
through Kratie (MRC, 2005). Figure 5 shows that water discharge varies from 1700 m$^3$/s to 40,000 m$^3$/s between dry and wet
seasons (Frappart et al., 2006; Le et al., 2007; MRC, 2009b; Wolanski et al., 1996). During the flood season high water

discharge causes inundation in the delta floodplains in Cambodia and Vietnam.

**Figure 5. Temporal distribution of daily water discharge at Kratie (available data from Darby et al., 2016).**

From Kratie to Phnom Penh, the hydrodynamics of the Mekong River are dominated by fluvial flows. The river banks are
lower than water levels in flood seasons, leading to water overflowing into the floodplains. The floodplains on the west side

convey water to the Tonle Sap River while the flood water flows into the Tole Touch River on the east side. The floodplains
on the west side receive less water than those on the east side, with water volumes of 24.7 and 35.4 km$^3$, respectively. The
peak discharges of the Mekong River to the left and right floodplains are approximately 5,400 and 7,800 m$^3$/s (Fujii et al.,
2003). These floodplains in combination with the Tonle Sap River cover about half of the Mekong's peak discharge.

At Phnom Penh, the Mekong is divided into two branches (Mekong and Bassac). In addition the Mekong River confluences

with the Tonle Sap River. The Tonle Sap Lake is the largest freshwater body in Southeast Asia and it has a crucially important
role in controlling water level in the Mekong inner part. Its surface area would cover an area of 3,500 km$^2$ during dry seasons
and is about four times larger during wet seasons (MRC, 2005). The water volume of the lake is up to 70 km$^3$ in the flood
season (MRC, 2005). The Tonle Sap Lake has a function as a natural flood retention basin of the Mekong River, leading to a
reduction of annual variations of water discharge flowing into the Delta. The flood flows to the Lake and reverses back during

low flows to the Mekong River at the Phnom Penh confluence. Figure 6 shows long-term daily average water discharge flowing
in and out of the Tonle Sap Lake at the Prek Kdam station. When water levels at Kampong Luong increase, reaching the peak
of over 9 meters, the Lake supplies water to the Delta, increasing Mekong River flows afterward and helping to reduce saline



intrusion in the coastal areas during dry seasons. From May to September, Mekong water feeds into the Tonle Sap Lake. From October until the following April it drains back into the Mekong.

**Figure 6. Daily averaged (from 1997 to 2004) water discharge hydrograph at Prek KDam and water level variation at Kampong Luong (Kummu et al., 2014). The solid line presents river flow coming in the Tonle Sap Lake at the Prek Kdam station, while the dashed line shows water levels at Kampong Luong station.**

From Phnom Penh to the Cambodia-Vietnam (CV) border, the Mekong River flows mainly through the Mekong branch, reaching up to 26,800 m³/s during flood peaks (Fujii et al., 2003). During the flood peaks, the floods discharge onto the VMD through the Mekong, Bassac branches and the floodplains overflow by 73%, 7% and 20% of the total discharge, respectively. In the VMD, the Mekong River flow diverts partly from the Song Tien (Mekong branch) to Song Hau (Bassac branch). Regarding flood distribution, water discharges at Tan Chau (Song Tien) and Chau Doc (Song Hau) are estimated to be 80% and 20% of the total flood flow, respectively. However, the connecting channel of Vam Nao leads to a relative balance between the Song Tien (at My Thuan) and the Song Hau (at Can Tho) downstream. At these stations water levels are strongly dominated by tides of the East Sea. Water levels in the coastal VMD fluctuate by tides from both the East Sea and West Sea, but the tidal range of the East Sea is much higher than that of the West Sea. Therefore, the East Sea's tides play a more important role and become the main dominant factor controlling hydrodynamics in the VMD coastal areas.

## 2 Methodology

### 2.1 Model description and setup

#### 2.1.1 Software description

The hydrodynamic model applied in this study is the Delft3D Flexible Mesh (DFM) Model Suite which has been developed by Deltares (deltares.nl). DFM is a multi-dimensional model which includes one, two and three dimensions in the same setup. It solves the two- and three-dimensional shallow water equations (Kernkamp et al., 2011). These equations describes mass and momentum conservation (Deltares, 2018).

$$\frac{\partial h}{\partial t} + \nabla.(h\boldsymbol{u}) = 0$$

$$\frac{\partial h\boldsymbol{u}}{\partial t} + \nabla.(h\boldsymbol{u}\boldsymbol{u}) = -gh\nabla\zeta + \nabla.\left(vh(\nabla\boldsymbol{u} + \nabla\boldsymbol{u}^T)\right) + \frac{\tau}{\rho}$$

Where $\nabla = \left(\frac{\partial}{\partial x}, \frac{\partial}{\partial y}\right)^T$, $\zeta$ is the water level, h the water depth, u the velocity vector, $g$ the gravitational acceleration, ν the viscosity, ρ the water mass density and τ is the bottom friction.

DFM allows computation on unstructured grids so it is suitable for regions with complex geometry (Achete et al., 2015), including combinations of 1D, 2D and 3D grids. This feature is efficient for taking into account small canals. Therefore, in





this study DFM is selected for simulating floods dynamics in the Mekong Delta which comprises a dense river network and highly variable river widths, dykes and flood plains.

### 2.1.2 Model setup

The VMD witnessed 3 large floods continuously from 2000 to 2002 based in the flood classification of the Tan Chau's flood peaks. Thus the 2000 and 2001 floods were chosen to calibrate and validate the model respectively. Another reason for selecting the 2000 flood is that datasets for this flood are comprehensive. The model in this study was improved from the model used by (Thanh et al., 2017). In the present configuration, the model uses a depth-averaged setting.

**Grid generation and improvement**

The unstructured model was constructed with an approach of multi-scale modelling; specifically, it consists of a combination of 1-D (canals) and 2-D (the main branches of the Mekong River, its floodplains and shelf) parts. The approach shows efficiency in the case of complex geometry such as the entire Mekong delta. To capture the hydrodynamics of the main branches and estuaries of the Delta, the main channels are represented in enough horizontal detail to resolve the flow patterns over channels and shoals and at the main bifurcations and confluences. Regarding the shelf, the model extended to approximately 80 km from the coastline of the Delta to fully contain the river plume (Figure 7).

The unstructured grid includes the river system of the Mekong River from Kratie to the East Sea and its shelf. The mainstream of the Mekong River, the subaqueous delta and floodplains are represented by 2D cells while the primary and secondary canals are modelled in 1D networks. The grid resolution varies from about 0.1 km in rivers to 3 km in the delta shelf. The lengths of grid are various depending on river geometry. The lengths of cells are generally around 700 m on the Mekong River mainstreams and reduce to approximately 200 m at river bifurcations and confluences. The larger cells of the Tonle Sap Lake, floodplains and Sea are up to 2000 m. The uniform length of 1D segments is 400 m. The grid quality is critical to accurate simulations, so the grid has been made orthogonal, smooth and sufficiently dense, to orthogonality values of less than 10%.

From the survey data, it can occur that a dyke ring in the model can consist of high dykes and low dykes together. This situation may occur because the model only includes the main rivers and secondary canal network, but tertiary and small canals are not included. In order to determine whether the dykes are fully protected or partly protected, the ratio of high dyke area and low dyke/non-dyke area is calculated. If the ratio is higher than or equal to 1, the dyke is recognised as a high dyke. If not, it is determined as a non-high dyke.

**Bathymetry data**

For modelling the flood dynamics in the Mekong Delta, bathymetry is a key element. However, available data of the Mekong Delta is limited. For river bathymetry, cross-sectional data has been used that was collected by the Mekong River Commission and used to develop the 1D hydrodynamic model (ISIS) to simulate fluvial flood propagation (Van et al., 2012). To use these profile data for 2D modelling, the cross-sectional data were interpolated to river bathymetry for the main branches while the primary and secondary canals directly used the data from the 1D network. For floodplains bathymetry, it is attached from





freely available digital elevation models of SRTM (Shuttle Radar Topography Mission). The bathymetry of the sea area is extracted from ETOPO at a resolution of approximately 1 km.

**Figure 7. River bathymetry from cross-section interpolation and shelf bathymetry of the Mekong Delta.**

**Boundary conditions**

Open boundaries are defined as water discharge (at Kratie) and water levels (the Sea). The measured water discharges were used for the upstream boundary at Kratie and were collected from the Mekong River Commission. The latter were defined as astronomical tidal constituents and extracted from a global tidal model (TPXO, Egbert and Erofeeva, 2002). Besides, in order to allow alongshore transport, the North cross-shore boundary is defined as Neumann boundary which is driven by the longshore water level gradient.

**Initial conditions**

Water levels in the Mekong Delta vary highly in space due to large-scale flood retention. The model takes a long time to capture the system behaviour, especially to arrive at the correct flood storage of the Tonle Sap Lake. The Tonle Sap Lake plays a significant role in controlling upstream discharge in the dry season. Therefore, the model was spun up over the year 1999; simulated results at the end of this year were used as initial conditions for the year 2000 simulation.

**2.2 Model calibration and validation**

The years of 2000 and 2001 were chosen for calibrating and validating the model respectively. The model calibration parameter is the roughness coefficient. This parameter is also selected for calibration without any sensitivity analysis since it is commonly used for calibrating hydrodynamic model (Manh et al., 2014; Wood et al., 2016). In this study, the 'trial and error' method is used for calibration. The roughness coefficients are extracted from the previous calibrated models, including ISIS (Van et al., 2012) and MIKE11 (Manh et al., 2014), in order to speed up the calibration process. The model was calibrated against measured data, with the objective function of Nash-Sutcliffe efficiency (NSE). NSE is a normalised statistical indicator that used comparison of residual variance and measured data variance (Nash and Sutcliffe, 1970) and calculated as:

$$E = 1 - \frac{\sum_{t=1}^{T}(Q_m^t - Q_o^t)^2}{\sum_{t=1}^{T}(Q_o^t - \bar{Q}_o)^2}$$

where $\overline{Q_o}$ is the mean of observed discharges, $Q_m^t$ is simulated discharges, and $Q_o^t$ is observed discharge at time t.

In this study, we used different temporal intervals of observation data. The daily data are used in the Cambodia Mekong Delta (CMD) and hourly in the VMD. The reason is that hydrodynamics in the CMD are unlikely affected by tides, particularly in flood seasons; while hydrodynamics in the VMD are strongly dominated by tides even in the flood seasons, so the hourly data are better for representing tidal fluctuation.

NSE is commonly used for evaluating hydrological models. Model performance is acceptable if NSE is higher than 0 (Moriasi et al., 2007). NSE is higher than 0; this mean the simulation is a better predictor than the mean observation. The NSE of 1




corresponds to a perfect match of modelled results to the observed data. The hydrodynamic model is defined as well calibrated if NSE in terms of water levels and discharges is higher than 0.5. Table 1 presents different categories of model performance based on NSE values (Moriasi et al., 2007).

**Table 1. NSE categories for evaluating model performance (Moriasi et al., 2007).**

In addition, we used an index of bias in order to recognize if the model has systematic under- or over estimates of water levels. In this study, a commonly used bias measure that is mean error is used to represent systematic error of the model (Walther and Moore, 2005). The bias is computed based on the following equation.

$$Bias = \bar{S} - \bar{O}$$

Where $\bar{S}$ is the simulated yearly mean and $\bar{O}$ is the observed yearly mean. The Bias is calculated for water levels over the year 2000.

## 2.3 High dyke development scenarios

To investigate the roles of different floodplains in the VMD and impact of these floodplains on the VMD's hydrodynamics and downstream tidal propagation, we developed scenarios that include contributions of each floodplains' water retention.

These scenarios used the hydrograph of the flood 2000. The hydrodynamic forcing is the same in these scenarios; the only difference is development of high dykes.

Scenario 1 (Base): This is the base scenario of the flood 2000, without high dykes.

Scenario 2 (Dyke 2011): Including the high dyke system in 2011 as illustrated in Figure 4.

Scenario 3 (Dyke LXQ): High dyke system developed only in the LXQ.

Scenario 4 (Dyke PoR): High dyke system developed only in the PoR.

Scenario 5 (Dyke TransBassac): High dyke system developed only in the Trans Bassac region. This region is a shallow inundated area.

Scenario 6 (Dyke VMD): High dyke system totally developed over the VMD's floodplains.

## 2.4 Analysis of simulations

### 2.4.1 Tidal harmonic analysis

The peak water level is a good index to indicate extreme events in the flooded areas. Tran et al. (2017) and Triet et al. (2017) used the flood peaks to assess the impact of high dykes in the VMD. However, the VMD coastal area is drastically dominated by tides. As a result, amplitudes of tidal constituents are good indices for presenting average variations of water levels in coastal areas. The water levels at the stations along the Song Tien and Song Hau were analysed over the whole year 2000 by

T_TIDE (Pawlowicz et al., 2002).



$$x(t) = b_0 + b_1 t + \sum_{k=1}^{N}(a_k e^{i\sigma_k t} + a_{-k}e^{-i\sigma_k t})$$

where N is a number of tidal constituents. We analysed the 8 main tidal constituents. Each constituent has a frequency $\sigma_k$ which is known, and a complex amplitude $a_k$ which is not known. $x(t)$ is a time series. $a_k$ and $a_{-k}$ are complex conjugates.

### 2.4.2 Water balance calculation

To understand flow dynamics, the water balance analysis is conducted by using hourly discharge data of simulations. The targeted stations for this analysis are located on the Mekong's mainstreams and boundaries of the flood-prone zones.

$$V_{in}^t = \sum_t Q * dt$$

where $V_{in}^t$ is total water volume flowing in the targeted regions in accordance with the Mekong flow's direction. $Q$ is hourly simulated discharge. $dt$ is the temporal interval. $t$ is selected periods of an entire year and seasons.

## 3  Results

### 3.1 Model calibration and validation

The overall model performance is good enough for simulating flood dynamics in the Mekong Delta. Figure 8 and Figure 9 show the NSE values of water levels and discharges respectively. For water level calibration, there are up to 36 stations used for calibration. Based on the mentioned classification, NSE at 33 of 36 stations are higher than the acceptable value of 0.5 in

which 23, 4 and 6 stations are classified as very good, good, and satisfactory categories, respectively. The 3 unsatisfactory stations are at Ca Mau, Phuoc Long and Rach Gia. Regarding to model validation, there are only 14 stations used for validation due to availability of observed data. Over the validation period, these stations have NSE values in the same groups with calibration. They are in good and very good classes, but the values at My Thuan stations are increased from 0.69 in calibration to 0.74 in validation.

**Figure 8. NSE values of water levels at the gauging stations in the Mekong Delta. The calibration (the year 2000) and validation (the year 2001) are presented on the left and right maps respectively.**

Table 2 presents calculated bias of water levels at stations in the Mekong Delta. Generally, the model slightly overestimates water levels. The large biases were found in the CMD, with the largest bias of around 1 m at Kratie. The absolute values of

biases decrease to smaller than 0.2 m at the stations in the VMD. Particularly, the biases at the middle and coastal VMD stations are smaller than 0.1 m.

**Table 2. Calculated bias for water level calibration at stations in different regions.**



The annual flood flows through the VMD by the Mekong mainstreams and over floodplains so discharge data of stations on these are employed in calibration. There are 11 stations on the mainstreams and across the CV border used in calibration. Nine out of 11 stations have NSE values in very good category. The two stations of across the border, namely Right Border (to the PoR) and Left Border (to the LXQ), are in satisfactory and unsatisfactory categories, respectively; however, the NSE values are 0.49 at Left Border and 0.54 at Right Border, fluctuating around the acceptable criteria. Data at these stations are not available for validation. All stations used for validation are in very good group. Compared to calibration, NSE values in validation are relatively stable, except My Thuan station which has an increase from 0.84 to 0.95. As a result, the Manning roughness coefficients of the Mekong River reaches and its floodplains after calibration and validation are illustrated in Table 3.

**Figure 9. NSE values of water discharges at the gauging stations in the Mekong Delta. The calibration (the year 2000) and validation (the year 2001) are presented on the left and right maps respectively.**

**Table 3. Calibrated values of Manning roughness coefficient.**

## 3.2 Spatial and temporal distribution of water volume in the VMD

### 3.2.1 Spatial distribution

Water enters the VMD by three ways: Song Tien, Song Hau and flows across the CV border. Figure 10 presents spatial distribution of water volume in the VMD. The VMD received around 580 km$^3$ in 2000, with volume of 405, 83, 61 and 31 km$^3$ through the Song Tien, Song Hau, the right and left CV border, respectively. The Song Tien diverts a considerable amount of 152 km$^3$ water to the Song Hau by the Vam Nao canal. This is the major mechanism to balance the flows seaward between the Song Tien and Song Hau. In fact, the streamflows are relatively equal between the Song Tien and Song Hau, with amounts of 247 (at My Thuan) and 235 km$^3$ (at Can Tho) respectively. The Song Tien is drained by its five estuary branches, while the Song Hau only has two branches. The Song Hau flows into the East Sea discharging 162 and 69 km$^3$ via the Dinh An and Tran De branches respectively. The Song Tien's estuary branches, namely Cung Hau, Co Chien, Ham Luong, Dai and Tieu, drain a similar volume to the East Sea, with a range of 54-63 km$^3$, except for the Tieu branch discharging only 34 km$^3$.

Besides the mainstreams of the Mekong River, floodplains have a substantial role in changing hydrodynamics in the VMD. Hence, we analysed the water balance on the three main flood-prone areas, consisting of the LXQ, PoR and TransBassac. Among of these, the PoR harbours the largest amount of floodwater, so it is a main flood storage of the VMD. Water flows into the PoR priminarily across the eastern part of the CV border in the Delta area. In fact, this way conveyed approximately 61 km$^3$ in 2000. The simulated results show a volume deficit of 29 km$^3$ from the West and South boundary of PoR that is drained to the Song Tien. The South of PoR drain a volume of around 38 km$^3$ to the Soai Rap estuarine branch by the Vamco River. Analysing the water balance of the LXQ shows that it receives water from the North and East sides, while it drains water to the West and South sides. The yearly inflow of the LXQ is about 44 km$^3$, with amounts of 31 and 13 km$^3$ from the



North and East boundaries, respectively. It is found that a similar amount of water drains out of the LXQ. The LXQ mainly releases water to the West bounday (32 km$^3$) into the West Sea, followed by the South boundary (13 km$^3$). The drained amount of 11 km$^3$ from the South LXQ mostly enters the TransBassac floodplains. An additional source into this TransBassac region is from the Song Hau, with yearly volume of 6 km$^3$. The sum of inflows is drained by the South canals of this region.

The principal dynamical characteristic of the Mekong Delta floods is their seasonal variation. Figure 10 and Figure 11 illustrate the seasonal variation of water volume and volume percent (compared to total entering volumes at Kratie) respectively. Obviously, the flows in flood season is significantly higher than those in the dry season. The flood flows contribute up to 53-65% of the annual flows throughout the mainstreams and the percentages increase to over 80% on the floodplains. The Mekong River flowing into the VMD in 2000 is about 97% of the total flow at Kratie. However, the water volume coming in the VMD

is higher than the entry volume at Kratie in the dry season.

In the dry season, there are slight discrepancies in the segments of the Mekong River. A part of the discrepant proportion is stored in the river segment. As evidence, water levels at Tan Chau at the beginning of the dry season is about 2 m, and increase to 3.5 m at the beginning of the flood season (Figure 13).

**Figure 10. Spatial distribution of water volume (km$^3$) throughout the VMD in 2000. The dry season is calculated from 01/Jan to 30/Jun and the flood season is from 01/Jul to 30/Oct.**

**Figure 11. Percentages (compared to total water volume at Kratie) of water distribution throughout the VMD in 2000.**

### 3.2.2 Temporal distribution

Figure 12 presents simulated fortnightly average discharges at in- and out-flows of the Mekong branches and cumulative water storage in the main VMD's floodplains from Apr-2000 to Apr-2001. The yearly average inflows of the Mekong branches at Tan Chau and Chau Doc are approximately 13,000 and 2,700 m$^3$/s, respectively. Playing a great role in water diversion between the two Mekong branches, the Vam Nao canal makes water discharges on the Song Tien and Song Hau more balanced seaward of the Vam Nao canal (Figure 10). Consequently, the water discharges at My Thuan (Song Tien) and Can Tho (Song Hau)

stations become almost similar, with annually average amounts of about 7,900 and 7,500 m$^3$/s respectively. The proportion at Can Tho station is simultaneously drained through the Song Hau mouths. The total outflow of the Song Tien is slightly greater than at My Thuan due to added flows from the southern PoR, discharging of around 8,400 m$^3$/s.

The water discharges on the Song Tien and Song Hau are highly variable over time. As shown for the results of discharge variations, the flood season is from the beginning of July to the end of October and the remaining period is defined as the dry

season. The largest seasonal difference is at Tan Chau, with the maximum and minimum discharges of about 21,000 and 4,500 m$^3$/s in the flood and dry seasons, respectively. The flood flow at Chau Doc reaches a peak of 5,600 m$^3$/s while the lowest flow is only 500 m$^3$/s in the dry season. However, the flood and dry flows on the Song Hau at Can Tho increase to over 14,100 and



2,200 m$^3$/s, respectively. A similar fluctuation is found at the Song Hau's mouths. On the Song Tien, the flood discharge at My Thuan is just 14,800 m$^3$/s and slightly rise to 17,000 m$^3$/s at the Song Tien mouths, but the dry flows are similarly of 2,400 m$^3$/s at these stations.

The hydrographs at the upstream VMD are flatter than downstream. The hydrograph shapes are indicated by their kurtoses and illustrated in Figure 12. The kurtosis index is a measure of the peakedness of the distribution. Downstream, the hydrographs are narrower at Can Tho, My Thuan and outflows of the Song Tien and Song Hau, with kurtosis higher than 1.5. One of the noticeable points is that during the flood season flows at Can Tho and My Thuan stations at the beginning are relatively lower than at the end, while the flood flows are stable throughout the flood season at Tan Chau and Chau Doc stations. This clearly shows how the early flood peak is stored in the major floodplains of the VMD. Figure 12 depicts the cumulative volumes in the major floodplains. At the beginning of the flood season, these floodplains are almost empty. By early October storage increases to 11, 8 and 2 km$^3$ in the PoR, LXQ and TranBassac, respectively. When these floodplains are filled, the flood flows at Can Tho and My Thuan reach their maxima during the year.

**Figure 12. Fortnightly average discharges at stations along the Mekong branches (left) and cumulative water volumes of the major floodplains in the VMD (right).**

### 3.3 Water level changes under high dyke development

Figure 13 shows that including high dykes increases daily mean water levels on Song Hau (Chau Doc, Long Xuyen and Can Tho) and on Song Tien (Tan Chau, Cao Lanh and My Thuan), especially in the flood season. The highest increase was found at Chau Doc and Tan Chau stations while increases decline more seaward.

### 3.3.1 Daily water levels

On the Song Hau the dyked floodplains in the LXQ, PoR and TransBassac cause increases of 12.3, 6.1 and 1.1 cm of annual mean water levels at Chau Doc station, respectively. However, Table 4 shows that the effect of the PoR dykes on water level at Long Xuyen and Can Tho is larger than that of the LXQ dykes. With the high dykes built until 2011, the yearly averaged water levels would increase by 10.2 cm (Chau Doc), 1.5 cm (Long Xuyen) and 0.2 cm (Can Tho). If the high dykes would be extended over the VMD (Scenario 6), the yearly mean water levels would rise up to 22.3 cm at Chau Doc station.

Generally, water levels on the Song Tien are less affected by the high dykes. Among the considered floodplains, the PoR has the highest effect on Song Tien's water levels since they are directly connected. For example, the yearly mean water level at Tan Chau increases by about 8.8 cm, but only 0.6 cm at Cao Lanh station. Interestingly, the PoR slightly reduces water levels at My Thuan due to reducing conveyed capacity of floodwater from the CV border. Although the LXQ is not directly linked to the Song Tien, it causes rising water levels by around 3.6 and 1.1 at Tan Chau and My Thuan, respectively. As the high dykes were covering 2,421 km$^2$ until 2011, the mean water levels are projected to rise by approximately 0.6 cm at My Thuan



and up to 6.1 cm at Tan Chau. In addition, the mean water level at Tan Chau could increase by 16.9 cm if the VMD's floodplains are fully dyked.

**Figure 13. Daily mean water level variations at selected stations along the Song Hau (left) and Song Tien (right) under different scenarios of high dyke development.**

**Table 4. Increases of yearly mean water levels (in cm) over the year 2000 at the selected stations along the Song Tien and Song Hau under different scenarios of high dyke development.**

### 3.3.2 Tidal amplitudes

The hydrodynamics in the Mekong Delta are significantly influenced by tides from the East Sea. A tidal harmonic analysis is conducted over the year 2000 to explore possible changes of the main tidal constituents. Figure 14 depicts the projected changes of tidal amplitudes along the Song Tien and Song Hau starting from the river mouths to approximately 195 km landward under high dyke development.

Tidal amplitudes at the river mouths are unlikely to change. However, differences become significant more inland. At Chau Doc, the LXQ causes the largest increase of tidal amplitudes compared to the other zones. It slightly increases M2 and K1 tidal amplitude by about 13% and 15% respectively. The TransBassac area has a main role in tidal amplitude change from Long Xuyen to Can Tho. Its dyked floodplains lead to an increase of the tidal amplitudes with 8 to 13%. Additionally, the M2 and K1 amplitudes could rise close to 28, 27 and 12% at Chau Doc, Long Xuyen and Can Tho, respectively. In contrast, high dykes in the PoR result in a marginal reduction of the amplitudes on the Song Hau. Similarly, the LXQ and TransBassac cause slight decreases of tidal variation on the Song Tien. High dykes constructed on the PoR leads to higher tidal amplitudes on the Song Tien, with increases of about 6%. These increases could reach to 28% at Tan Chau, 11% at Cao Lanh and 4.4% at My Thuan.

**Figure 14. Tidal amplitudes of the 8 main constituents at the selected stations along the Song Tien (right) and Song Hau (left) from the river mouths to about 195 km landward in the scenarios of high dyke development.**

## 4 Discussion

### 4.1 Model performance

The calibration presented in this study considered a larger number of stations compared to previous studies. These stations are mainly located in the VMD (Figure 8). The majority of stations on the main Mekong River's branches have a NSE higher than 0.8 (good). In contrast, the stations located further away from the main stream have a lower NSE. The NSE values of Phuoc Long and Ca Mau stations are lower than the acceptable level because the water levels at these stations are highly influenced





by local infrastructure, specifically the Quan Lo Phung Hiep Project (QLPH). The QLPH is built to protect this area from saline intrusion. Flows entering QLPH are controlled by a series of sluice gates mainly located along the coast to avoid saline intrusion to areas for rice cultivation and control fresh water sources.  We did not consider these sluice gates in the model, because they do not have a fixed operation schedule, but one that is based on crop calendars and in-situ hydrodynamics (Manh

et al., 2014). For example, observed water level at Phuoc Long station is relatively unchanged at 0.2 m over the year 2000, while the model estimates that water levels at this station have semi-diurnal variations between -0.2 m and 0.6 due to tidal effects of the East Sea. During validation, a better fit was found at My Thuan station, while the others stations have comparable NSE values. As such, we are confident that the model is capable of capturing hydrodynamics in the Mekong Delta accurately.

## 4.2 Spatio-temporal distribution of water volume in the VMD

The total net water volume flow through the Mekong Delta at Kratie is approximately 600 km$^3$ in 2000 where the annual flood contributed about 480 km$^3$. This is considerably higher than the average volume of 330 km$^3$. However, the annual flood peak in 2000 is just slightly higher than the mean flood peak of 52.000 m$^3$/s (MRC, 2009a). Thus, the 2000 flood is characterized by a broader than usual hydrograph.

Several studies have investigated the distribution of flood volume in the Mekong Delta (e.g. Manh et al. (2014), Nguyen et al.

(2008) and Renaud and Kuenzer (2012)). Manh et al. (2014) calculated flood volume distribution for the floods between 2009 and 2011 in the upper VMD and concluded that the flood distribution is marginally changed over the mentioned period. However, they did not estimate flow distribution through the river mouths. We found a similar pattern of flood volume distribution on the mainstreams, but our model estimated a larger discharge across the VC border to the VMD. A possible explanation is that the 2000 flood is considerably larger than the floods during the 2009-2011 period. Table 5 shows a

comparison of the VMD's outflows of the current study and five other model as summarized by Nguyen et al. (2008). There is only a small variation among the used models which is attributed to different topographical data and boundary conditions (Nguyen et al., 2008). The flow distribution of the current study falls within the range of variation of the other five models, though it differs in some important branches such as Song Tien and Song Hau, below Vam Nao.

**Table 5. Distribution of water discharge throughout the river mouths (after Nguyen et al., 2008).**

The water distributions slightly vary over the flood and dry seasons. The largest changes are found in the discharges onto the floodplains. For instance, water volumes are highly seasonal at the CV border stations. The water flows in the dry season contribute to 2 - 6% of the yearly flows at these stations. The relative percentages of the Mekong flow, existing via the Song Hau estuaries in the dry season are higher than those in the flood season while the percentages at the Song Tien estuaries are

relatively constant.

There are several studies which investigate the roles of the Tonle Sap Lake in regulating the flood regimes on the Mekong River (Fujii et al., 2003; Kummu et al., 2014; Manh et al., 2014). Kummu et al. (2014) estimated that the Tonle Sap Lake is



capable of reducing about 20% of the Mekong mainstream discharge and its greatest storing volume is in August, with an amount of around 15 km3 from the Mekong River flows. The highest monthly released volume occurs in November and peaks at nearly 20 km$^3$. Consequently, the Tonle Sap Lake has a crucial role on the seasonal temporal scale in regulating the Mekong River flows. The VMD floodplains have a different role in changing the Mekong mainstream flows. They mainly store early

flood waters in August. This leads to reduce flood flows at downstream stations along these floodplains. These stations reach the peak discharges when the VMD floodplains are nearly fully filled. Therefore, the peak flows at the downstream stations occur in October. These results are consistent with the analysis of Dang et al. (2018).

### 4.3 Impact of high dyke development

There are several threats to the current status of the Mekong Delta, such as the impact of hydropower dams, sea level rise,

delta land subsidence, and hydraulic infrastructure (Duc et al., 2018). Impact of these threats highly depends on temporal scales. Among of these, hydraulic infrastructure (especially high dykes) has considerable influence on the hydrodynamics in the region on a short temporal scale. The high dykes in the VMD are built to protect agricultural land during floods. As a result, flood discharges on the rivers increase and hydrodynamics in the VMD change. Specifically, the results indicate that lack of flood retention in the LXQ leads to an increase of water levels on the Song Hau, with a downward trend of increases from

Chau Doc to Can Tho. This rising pattern is found by Tran et al. (2017) as well, albeit with different magnitudes because of different comparison. They compared the peak water levels while we used daily mean water levels for comparison. Tran et al. (2017) found that the water level peaks would be drastically higher if the high dykes were built. These peaks have specially increased in the upper VMD (e.g. by 66 cm at Chau Doc and only 4 cm at Can Tho).

Interestingly, the high dykes in the PoR has slightly stronger impacts on water levels at Long Xuyen station than those in the

LXQ. The reason for that is an increase of water levels on the Song Tien, causing an increase of water diversion from the Song Tien to the Song Hau. Because of the connecting canal of Vam Nao, the PoR floodplains have not only influenced water level fluctuation on the Song Tien, but also on the Song Hau. In addition, the LXQ floodplains affect both the Song Tien and Song Hau water levels. Nevertheless, the increasing levels on the Song Tien remain slightly lower than the levels on the Song Hau since the Song Tien has more river mouths and has a higher conveyance capacity in comparison to the Song Hau.

Recent studies on impact of high dykes in the VMD (e.g. Tran et al. (2017) and Triet et al. (2017)) only compared the maximum water levels. However, we found that the high dykes also resulted in reduction of minimum water levels. This mean the high dykes have effects on tidal fluctuation on the main branches. We analysed tidal amplitudes of the 8 main constituents over the year 2000 in order to quantify how water levels on the main branches changed. Noticeably, the complete implementation of the high dyke system over the VMD floodplains can cause increases of about 12% and 4% tidal amplitudes at Can Tho and

My Thuan stations, respectively. Additionally, high dykes in the PoR directly adjacent to Song Tien cause tidal amplitude reduction on Song Hau and vice versa. The reason is that river water cannot flow into the floodplains, leading to an increase of river discharge in the main streams. This increased river discharge causes significant M2 amplitude reduction (Guo et al.,



2016). The amplitudes and mean water level at the river mouth stations are unlikely to change under high dyke development (Table 4). The reason is that flood retention loss due to embankments has insignificant changes on the water discharge at that location. In contrast and as an example, Kuang et al. (2017) found that if the water discharges from the Yangtze River upstream would increase by 20,000 m³/s, the sea levels at the mouth could rise approximately 1 cm. An explanation is that the water

discharge change due to high dyke development is not larger enough to increase water levels at the river mouths.

## 5 Conclusions

In this study, we applied a process-based model (DFM) in order to simulate hydrodynamics in the entire Mekong Delta from Kratie to extended sea areas. The model was calibrated by a comprehensive dataset of water levels and discharge over the Mekong Delta. As a result, the model shows a good agreement between simulations and observations. This model is an

improved version of the model used by Thanh et al. (2017), by taking into account the Cambodian and Vietnamese floodplains and the dense river/canal network in the VMD. Nevertheless, it does not contain tertiary rivers/canals and hydraulic structures for salinity regulation.

We found that the change in river flow distribution throughout the Mekong's mouths is insignificant, except at the Dinh An mouth which has a slight increase in the dry season. In contrast, the Mekong River network discharging to the sea through the

Soai Rap mouth and the West LXQ dramatically dropped in the dry season compared to the flood season due to overflow reduction at the CV border.

This study found that the dyked floodplains in the LXQ and PoR not only influence water regimes on its directly linked Mekong' branch, but also on the other branches. The LXQ high dykes cause an increase in daily mean water levels, but a decrease in tidal amplitudes on the Song Tien (after the connecting channel of Vam Nao). A similar pattern is also found for

the interaction between the PoR high dykes and the Song Hau. The high dykes built in the PoR, LXQ and TransBassac regions have a demonstrated impact on water levels at Tan Chau, Chau Doc and Can Tho, respectively.

## Appendix

Scatter plots of simulation and measurement in terms of water levels and flow discharges.





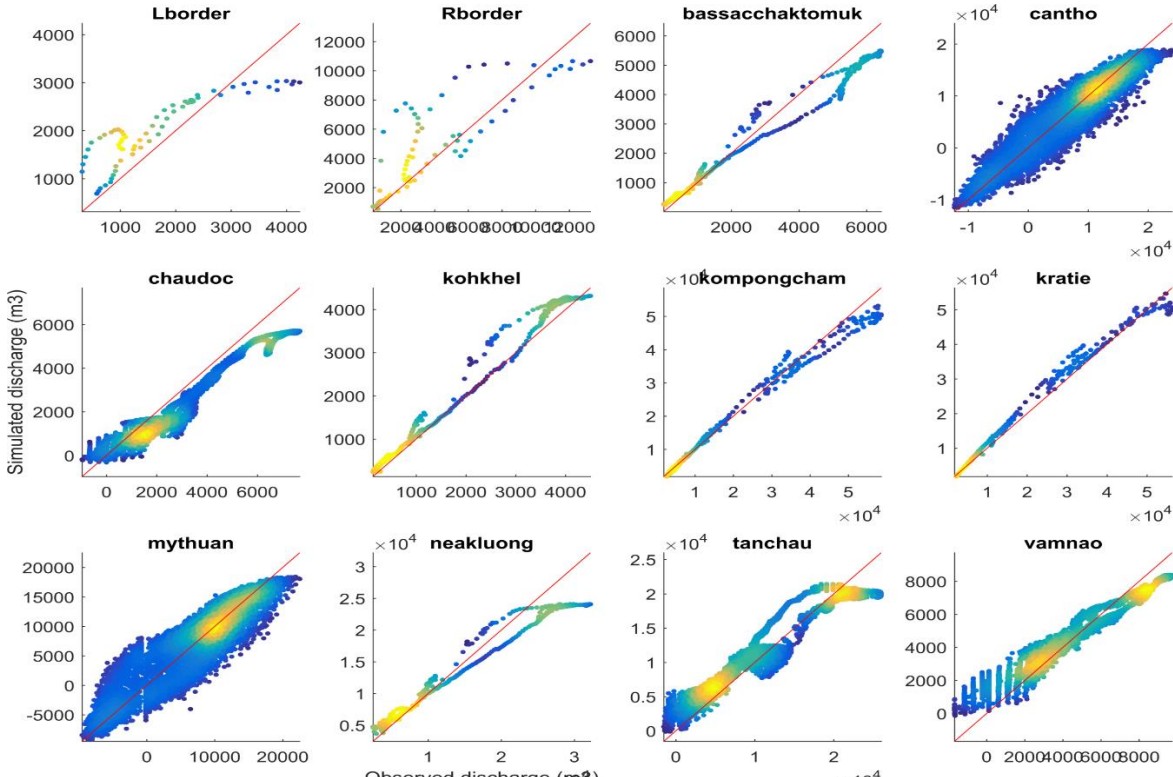

**Figure A. Scatter plots of simulated and measured discharges.**



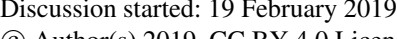

**Figure B. Scatter plots of simulated and measured water levels.**

*Acknowledgements*

This project is part of the ONR Tropical Deltas DRI and is funded under grants N00014-12-1-0433 and N00014-15-1-2824.

5   The authors would like to thank Dr. Tran Duc Dung and the Mekong River Commission for providing the data. Simulations were carried out on the Dutch national e-infrastructure with the support of the SURF Cooperative.

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

**Tables**

**Table 1. NSE categories for evaluating model performance (Moriasi et al., 2007).**

| Performance | NSE ranges |
|---|---|
| Unsatisfactory | NSE ≤ 0.5 |
| Satisfactory | 0.65 ≤ NSE ≤ 0.5 |
| Good | 0.75 ≤ NSE ≤ 0.65 |
| Very good | 1 ≤ NSE ≤ 0.75 |

**Table 2. Calculated bias for water level calibration at stations in different regions.**

| CMD | Bias (m) | Upstream VMD | Bias (m) | Middle VMD | Bias (m) | Coastal VMD | Bias (m) | Coastal VMD | Bias (m) |
|---|---|---|---|---|---|---|---|---|---|
| Kratie | 1.02 | Tan Chau | 0.12 | Kien Binh | -0.03 | An Thuan | -0.02 | Nam Can | -0.24 |
| Kampong Cham | 0.54 | Chau Doc | -0.20 | My Thuan | -0.08 | Ben Trai | -0.04 | Phung Hiep | -0.07 |
| Prekkdam | 0.92 | Vam Nao | -0.10 | Can Tho | 0.02 | Binh Dai | 0.00 | Phuoc Long | -0.06 |
| Chaktomuk | -0.15 | Xuan To | 0.09 | Hung Thanh | 0.00 | Ca Mau | -0.18 | Rach Gia | -0.08 |



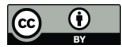

| Kohkhel | -0.51 | Moc Hoa | 0.04 | Tuyen Nhon | 0.06 | Dai Ngai | -0.02 | Song Doc | -0.03 |
|---|---|---|---|---|---|---|---|---|---|
| | | Long Xuyen | -0.12 | | | Ganh Hao | -0.07 | Tan An | 0.08 |
| | | | | | | Hoa Binh | 0.09 | Tan Hiep | -0.10 |
| | | | | | | My Hoa | 0.00 | Tra Vinh | 0.05 |
| | | | | | | My Thanh | 0.02 | Vam Kenh | 0.07 |
| | | | | | | My Tho | 0.02 | Vi Thanh | 0.00 |

**Table 3. Calibrated values of Manning roughness coefficient.**

| River reaches/floodplains | Manning roughness coefficient | River reaches/floodplains | Manning roughness coefficient |
|---|---|---|---|
| Kratie to Phnom Penh | 0.031 | The Tonle Sap Lake and River | 0.032 |
| Cambodia floodplains | 0.036 | Phnom Penh to Vam Nao (Song Hau) | 0.033 |
| Phnom Penh to Tan Chau | 0.031 | Vam Nao to Can Tho (Song Hau) | 0.027 |
| Tan Chau to My Thuan | 0.029 | Can Tho to Song Hau mouths | 0.021 |
| VMD floodplains | 0.018 | VMD channels | 0.027 |
| My Thuan to Song Tien mouths | 0.023 | Continental shelf | 0.016 |

**Table 4. Increases of yearly mean water levels (in cm) over the year 2000 at the selected stations along the Song Tien**

5  **and Song Hau under different scenarios of high dyke development.**

| Station Scenario | Song Hau | | | | Song Tien | | | |
|---|---|---|---|---|---|---|---|---|
| | Chaudoc (cm) | Longxuyen (cm) | Cantho (cm) | Dinhan (cm) | Tanchau (cm) | Caolanh (cm) | Mythuan (cm) | Bentrai (cm) |
| Dyke 2011 | 10.2 | 1.5 | 0.2 | 5.0x10-3 | 6.1 | 3.4 | 0.6 | 1.8x10-2 |
| Dyke LXQ | 12.3 | 3.2 | 1.0 | 1.4 x10-2 | 3.6 | 2.6 | 1.1 | 2.8 x10-2 |
| Dyke PoR | 6.1 | 3.6 | 1.2 | 1.5 x10-2 | 8.8 | 0.6 | -0.8 | -1.9 x10-2 |
| Dyke TransBassac | 1.1 | 1.9 | 0.7 | 1.7 x10-2 | 0.8 | 0.7 | 0.3 | 1.4 x10-2 |
| Dyke VMD | 22.3 | 9.2 | 3.0 | 4.6 x10-2 | 16.9 | 6.6 | 1.4 | 4.6 x10-2 |

**Table 5. Distribution of water discharge throughout the river mouths (after Nguyen et al., 2008).**





| Model name | The Song Tien below Vam Nao (%) | The Song Hau below Vam Nao (%) | Co Chien (%) | Cung Hau (%) | Dinh An (%) | Tran De (%) | Ba Lai (%) | Ham Luong (%) | Tieu (%) | Dai (%) | Others (%) |
|---|---|---|---|---|---|---|---|---|---|---|---|
| NEDECO 1974 | 51 | 49 | 13 | 15 | 28 | 21 | 0 | 15 | 2 | 6 | 0 |
| VNHS 1984 | 55 | 45 | 13 | 18 | 27 | 18 | 0 | 17 | 1 | 6 | 0 |
| SALO89 1991 | 44 | 54 | 12 | 8 | 26 | 24 | 2 | 14 | 5 | 2 | 8 |
| Nguyen Van So 1992 | – | – | 11 | 12 | 19 | 16 | 1 | 14 | 1.5 | 6 | 20 |
| VRSAP 1993 | 50 | 44 | 11 | 5 | 18 | 18 | 0 | 9 | 2 | 8 | 29 |
| This study | 41 | 39 | 10 | 11 | 27 | 12 | 0 | 9 | 6 | 9 | 14 |

Note: in this study the percentages are calculated based on the total volume at Kratie.




**Figures**

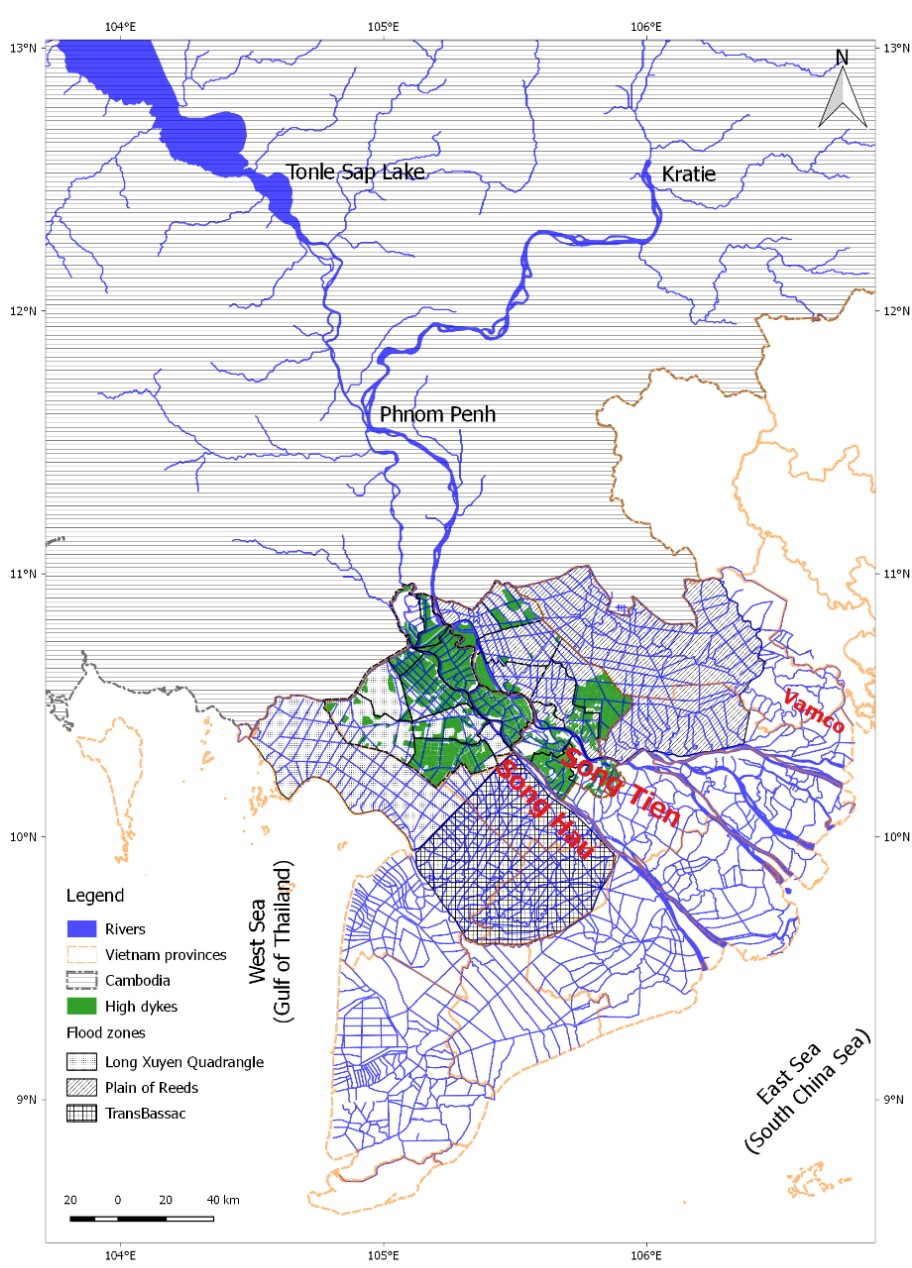

**Figure 1. Location of the Mekong Delta.**



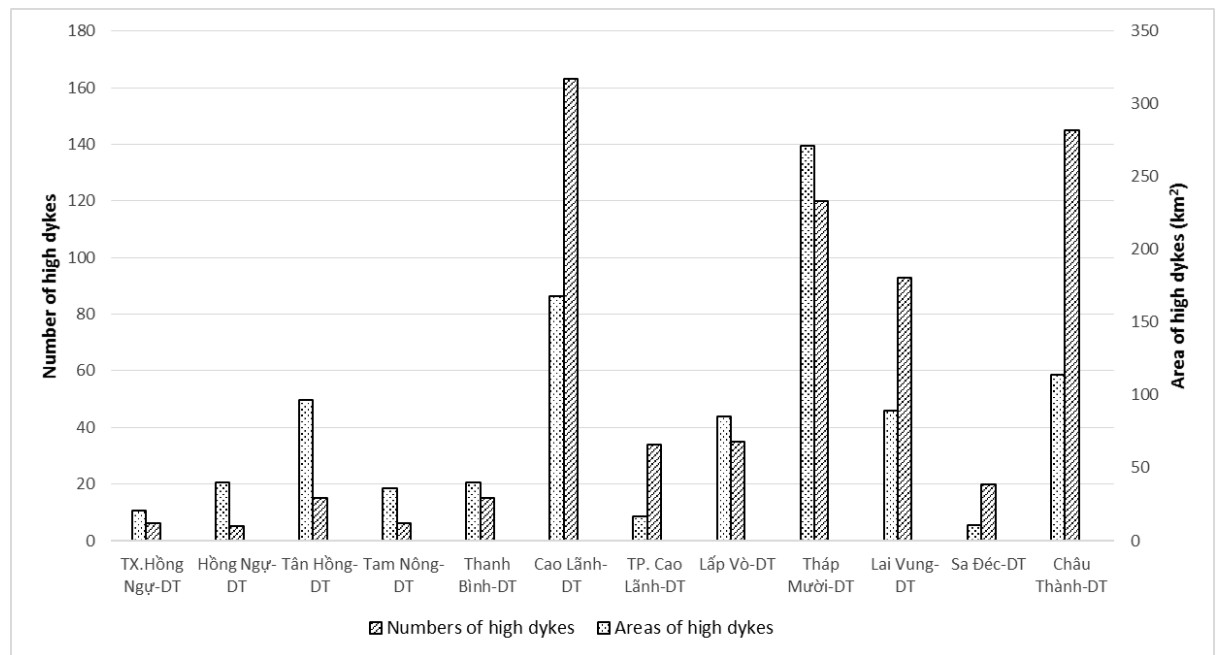

**Figure 2. Numbers of high dykes and its areas by districts in Dong Thap province.**

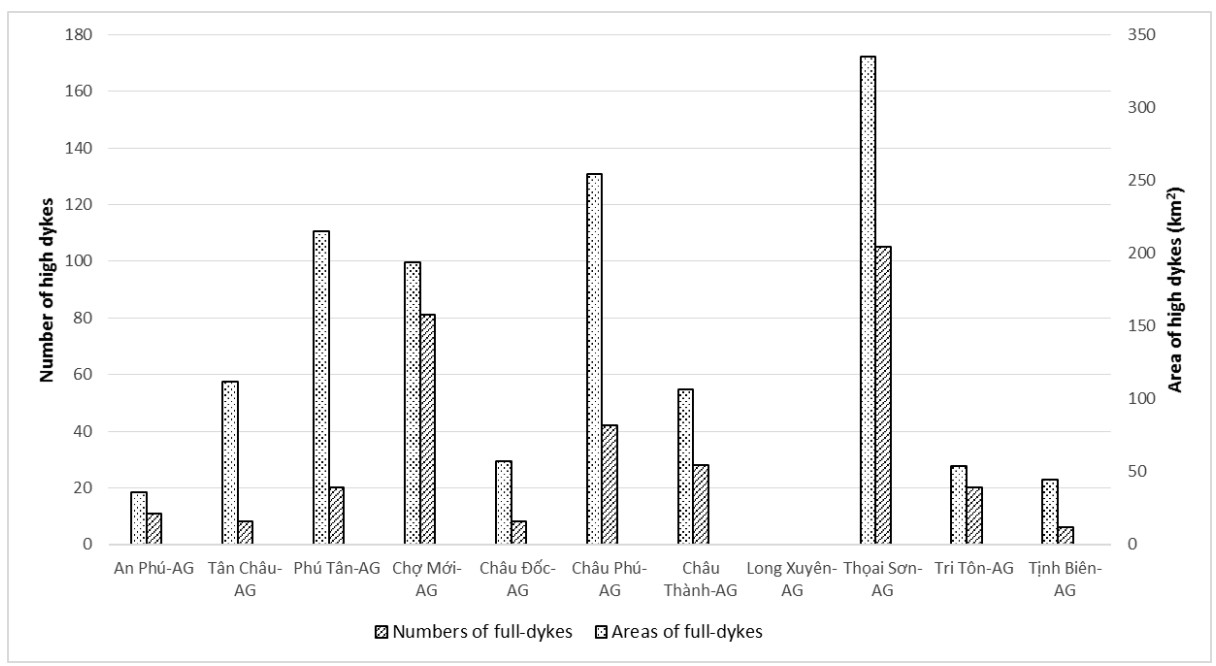

**Figure 3. Numbers of high dykes and its areas by districts in An Giang province.**





**Figure 4. Spatial distribution of high dykes which were built until 2011.**





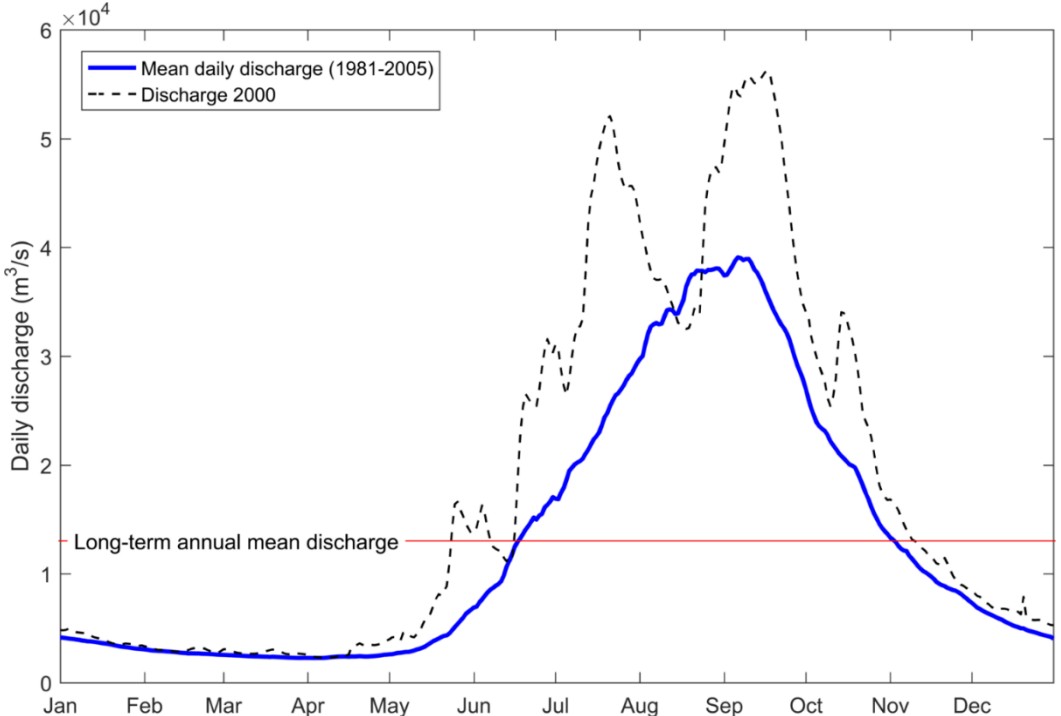

**Figure 5. Temporal distribution of daily water discharge at Kratie (available data from Darby et al., 2016).**

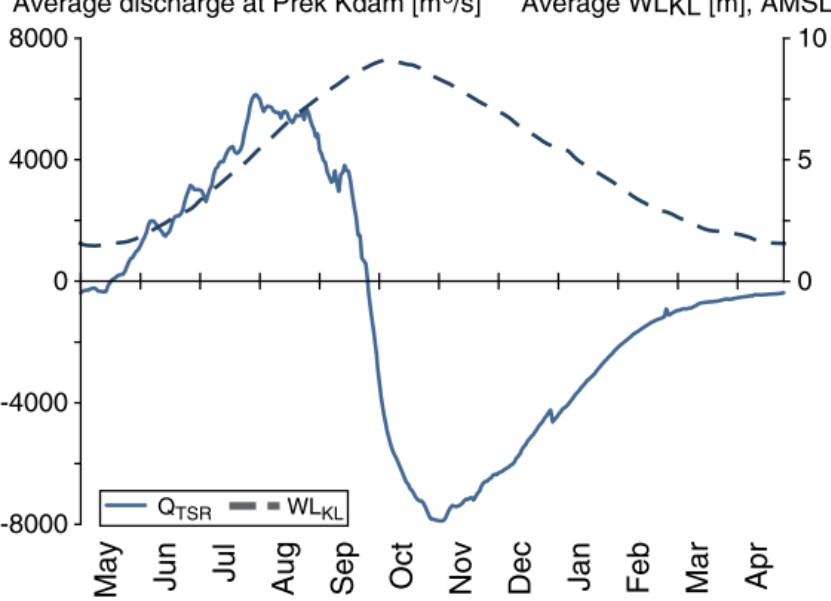

**Figure 6. Daily averaged (from 1997 to 2004) water discharge hydrograph at Prek KDam and water level variation at**

5   **Kampong Luong (Kummu et al., 2014). The solid line presents river flow coming in the Tonle Sap Lake at the Prek**

  **Kdam station, while the dashed line shows water levels at Kampong Luong station.**





**Figure 7. River bathymetry from cross-section interpolation and shelf bathymetry of the Mekong Delta.**





**Figure 8. NSE values of water levels at the gauging stations in the Mekong Delta. The calibration (the year 2000) and validation (the year 2001) are presented on the left and right maps respectively.**



**Figure 9. NSE values of water discharges at the gauging stations in the Mekong Delta. The calibration (the year 2000) and validation (the year 2001) are presented on the left and right maps respectively.**



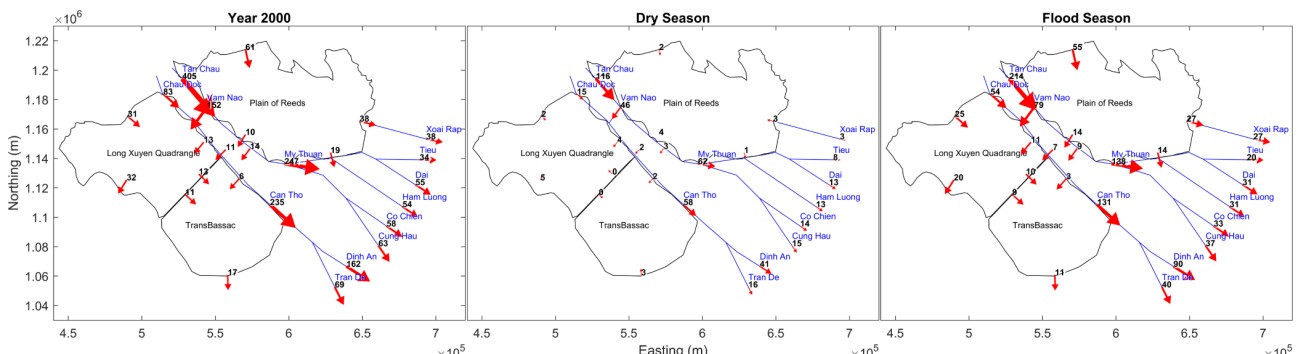

**Figure 10. Spatial distribution of water volume (km3) throughout the VMD in 2000. The dry season is calculated from 01/Jan to 30/Jun and the flood season is from 01/Jul to 30/Oct.**

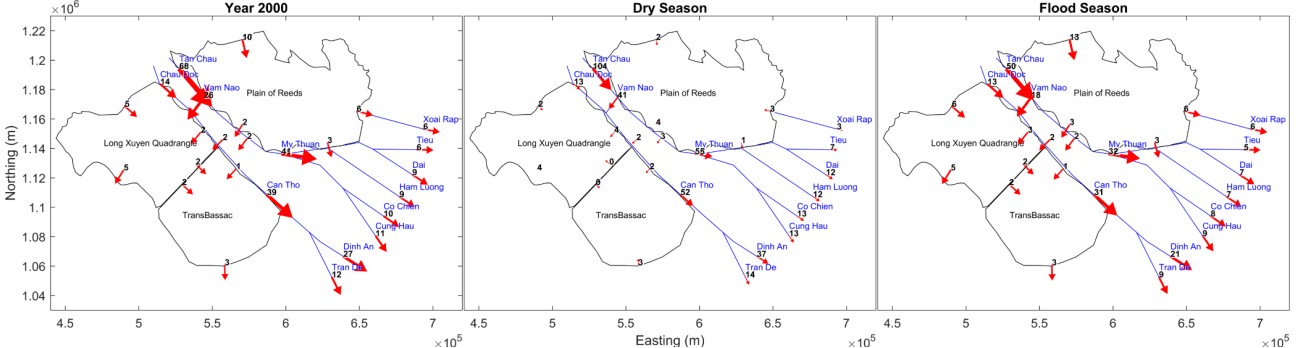

**Figure 11. Percentages (compared to total water volume at Kratie) of water distribution throughout the VMD in 2000.**





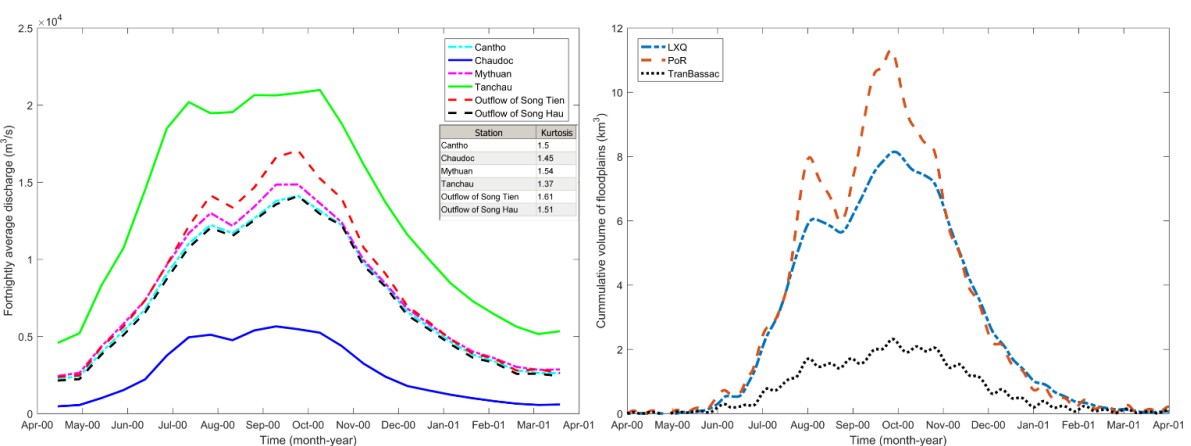

**Figure 12. Fortnightly average discharges at stations along the Mekong branches (left) and cumulative water volumes of the major floodplains in the VMD (right).**

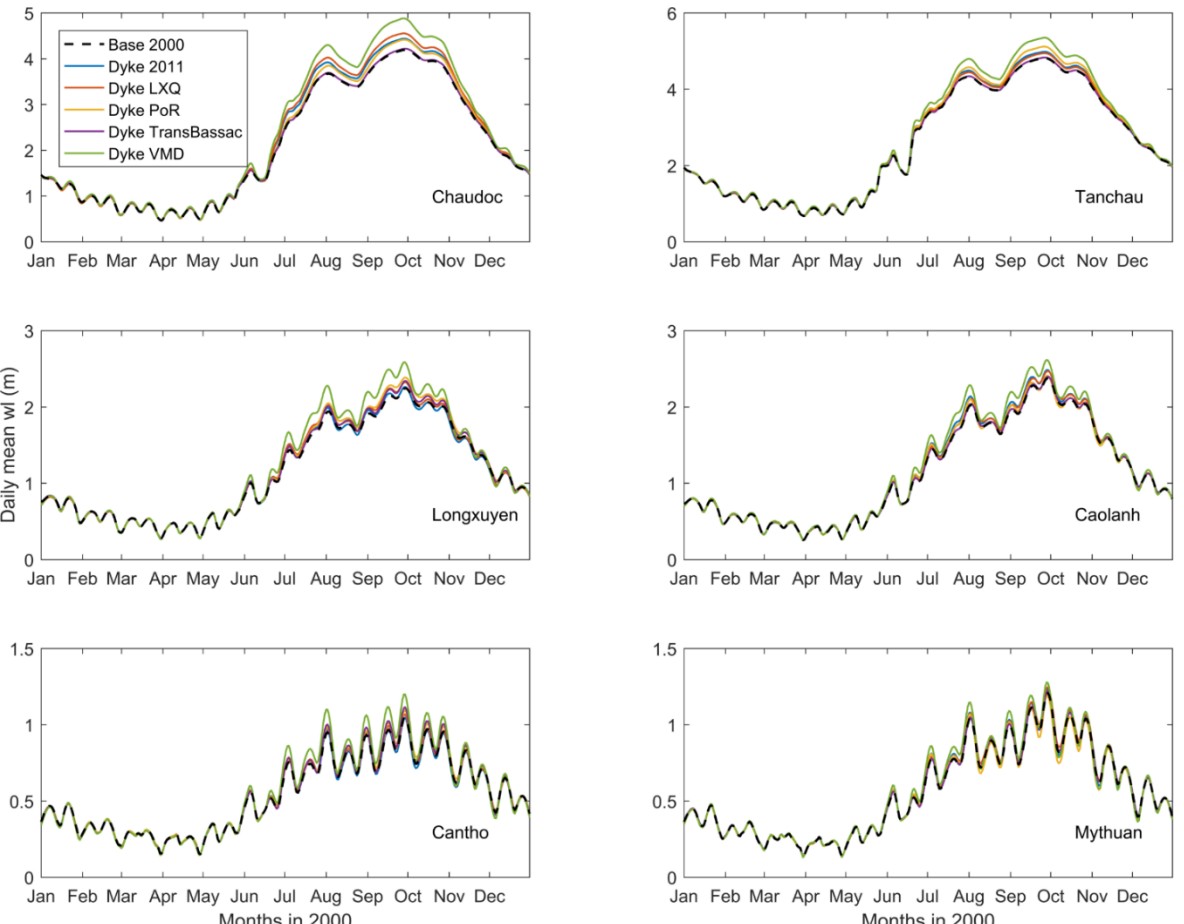

5  **Figure 13. Daily mean water level variations at selected stations along the Song Hau (left) and Song Tien (right) under different scenarios of high dyke development.**

**Figure 14. Tidal amplitudes of the 8 main constituents at the selected stations along the Song Tien (right) and Song Hau (left) from the river mouths to about 195 km landward in the scenarios of high dyke development.**