# Peer review of "Flooding in the Mekong Delta: Impact of dyke systems on downstream hydrodynamics"

_Hydrology and Earth System Sciences, 2019_

## Referee Comment (RC1) · Anonymous Referee #1 · 28 Mar 2019

The authors use a 1D/2D hydrodynamic model that covers the Mekong Delta including its rivers, major canals and extending into the continental shelf in the surrounding ocean, to investigate the impact of protecting agricultural areas with high-dykes on the river hydrodynamics. They found that (a) High dykes (particularly those in Long Xuyen Quadrangle (LXQ), Plains of Reeds (PoR)) have wide-spread impact on the flow downstream (b) has impact on inland tidal effects. Recent literature is well covered. Reasonably well-written introduction. Language use is generally adequate (though there are a number of technical issues that need correcting.) However, the paper is not easy to read as it is organized in such a way that a lot of (seemingly unnecessary) material is mixed with the main narrative of the paper. Following are some of the major issues:

1. What is the benefit of modelling the continental shelf? This is not an oceanographic/coastal engineering study. Your focus (as stated) was to investigate the impact of construction of high-dykes on the flow regime of the river system. You are also not considering highly dynamic ocean impacts like storm surge. What is the drawback of stopping the model at the river mouth and providing tidal boundary conditions with sea level there? You might have good reasons for this approach. If so, they need to be explained.

2. One year of simulation is a short period to obtain meaningful results. I think it is important to cover at least several years of flow data as such data for this case study is available. Is there a barrier to doing that?

3. The point of departure (and justification for the methodology) of this paper seems to be the fact that previous studies could not able to predict the water level at the river mouth. If this is the sole justification to use a numerically expensive 2D model that includes continental shelf, the importance of obtaining those figures should be explained.

4. The point of doing a tidal harmonic analysis is unclear to me. Just testing the impact on the tidal range (amplitude) would have covered all the matter that is relevant to the central theme of the paper. Removing the tidal harmonic analysis part would shorten the paper - definitely would contribute to making it more readable and to the point.

5. The authors should discuss the performance of the model. This is particularly important as many previous studies have used (much simpler) 1D modelling approach to arrive at similar results. How much is the computational effort? How does it compare with those reported in previous studies? What is the justification to use this modelling approach despite its expense (if that is the case)?

6. So many figures and many descriptions on model validation performance. This is an important topic to cover, but it is overdone in this case. Just one paragraph on

how the model performed during validation and if absolutely necessary, one map showing validation results. Much of this can be moved to an appendix. In fact, it's best that they are presented as an online supplement rather than an Appendix, so as to keep the paper succinct and to the point.

7. Scenarios need a better explanation. For example how much is protected with high dykes in "Dyke VMD" scenario? What is the basis?

8. Water balance diagrams and descriptions are hard to understand. Please check the literature for much clearer ways of presenting these.

9. Lastly, it is important to place the findings within the context of other changes. Are these significant for example impact of climate change on upstream Mekong flow, dam construction, sea-level rise etc.? Some discussion on such issues is warranted.

---

## Referee Comment (RC2) · Anonymous Referee #1 · 4 Apr 2019

**Flooding in the Mekong Delta: Impact of dyke systems on downstream hydrodynamics**

Vo Quoc Thanh1,2,3, Dano Roelvink1,2,4, Mick van der Wegen1,4, Johan Reyns1,4, Herman Kernkamp4, Giap Van Vinh5, Vo Thi Phuong Linh3

[referee-annotated manuscript omitted]

---

## Referee Comment (RC3) · Anonymous Referee #2 · 12 Apr 2019

**Review of manuscript:** "Flooding in the Mekong Delta: Impact of dyke systems on downstream hydrodynamics*".*

**Overview**

The paper describes the effects of the high dykes structures on the complex interaction of the high flows and tides in Mekong Delta. The authors use 1D-2D model Delft3d Flexible Mesh model on unstructured grid to simulate the hydrodynamics at the river branches, canals and the adjacent ocean shelf with certain improvements compared to the previous studies done on Mekong Delta.

No major concerns about the language use, however the manuscript would benefit from the proofread by a native speaker. The research question is clearly stated and addressed in the Discussion section.

**The main concerns**

The overall structure of the manuscript is sufficient, however the abundance of details (some of those are unnecessary in my opinion) makes it difficult to read. The modelling part of the study needs major revisions (see comments below). Moreover, the figures should be revised. Therefore, I suggest major revisions of content further on.

**General comments**

Introduction. This section is somewhat well-structured and clear, however some information which is repetitive or not directly related to the study objective should be removed. For instance, speaking of the soil quality or challenges in agriculture may confuse the reader in the beginning. One sentence should be enough for it.

Methods. The following aspects need to be addressed:

- Input data. The authors use SRTM data to run the 2D part of the model. SRTM is known to have rather large vertical and horizontal errors, however, there is no overview related to the data pre-processing, resolution used and possible errors coming from the input data accuracy. Some of the simulated water level changes are far smaller than the vertical accuracies of the terrain and bathymetry data. This aspect should be deliberately described in the Methods and Discussion section. The following paper might be useful: Hawker, L. P., Rougier, J., Neal, J. C., Bates, P. D., Archer, L., & Yamazaki, D. (2018). Implications of simulating global digital elevation models for flood inundation studies. Water Resources Research, 54. https://doi.org/10.1029/2018WR023279.
- High dyke geometry should be described (design return period, crest elevation. precise location etc.) and how they were incorporated into the computational mesh.
- Simulation time, if possible, should be increased to more years. One year is not sufficient enough to track the system behaviour in given setting and may cause biases.
- The authors mention and illustrate throughout the manuscript high dyke as both, a flood protection measure and a dyke protected floodplain at the same time. This causes ambiguity and should be further specified/changed.
- More details about the developed scenarios need to be added (see the dyke geometry).

Results. The calibration/validation outcomes description should be substantially reduced, as it overloads the section. Computational time should be mentioned.

Discussion. I would recommend restructuring the discussion by answering the research questions in the same order as outlined in the Introduction section. It should be pointed out how the study benefited

from using 2D code compared to 1D (MIKE11 or ISIS). The limitations related to data and methods used have to be put in a separate sub-section.

Conclusion. Implications and future work should be added.

Figures. Geographical names and symbols should be consistent throughout the manuscript.

**Specific remarks**

P.2 line 17 – the sea level rise and land subsidence is an important point in the study area, the data about the future projections can be mentioned. However, it is important to point out why they are not considered in the paper (complexity, uncertainty, etc.).

P.3 line 4 – sentence "These outcomes… " belongs to Conclusion

P.3 line 15- Figure 4 comes right after Figure 1

P.4 line 3 – the reference seems to be outdated

P.4 line 12 – return period of the event should be specified

P.4 line 14 – sentence "The water…" is ambiguous

P.4 line 31 – units should be specified

P.7 line 21 - the grid density is mentioned to be sufficient, however the grid cell sizes seem to be quite large in fact. It would be useful to have some more explanation for the readers who are not familiar with Delft3DFM model

P.7 line 32 – floodplain  topography

Table 1 can be removed

Figure 1. The map is hard to read. I would recommend changing the following: reduce thickness of canal network, make the flood zones boundaries and fill more pronounced. The outline of municipalities (the faded orange line) should be removed. It is better to avoid overlap of green and red colours if possible

Figure 2 and 3. What is meant here by high dykes? Dyke protected floodplains or flood defence? Should be clearly stated. Is there a way to combine two figures in one or incorporate in the Figure 4?

Figure 4. One of the north arrows should be removed. Region boundaries are not visible

Figure 7. Everything that is above the sea-level (0m a.s.l.) is topography

Figure 8 and 9. In legend the sign ">=" should be corrected

Figure 10 and 11. The legend which specified the thickness of red arrows should be added

---

## Referee Comment (RC4) · Anonymous Referee #3 · 14 Apr 2019

Review of manuscript: "Flooding in the Mekong Delta: Impact of dyke systems on downstream hydrodynamics"

The study aims at evaluating the impact of high dykes on the hydrodynamic behavior of the Mekong river. The analysis is based on a 1D-2D model that reproduces the topographic characteristics of the river, as well as different dyke ring configurations. The work does not infer specific research questions but it is aimed at increasing the knowledge of the river dynamics and its behavior in the light of the recent construction of a complex dyke system. The manuscript is in general well written, even if it results sometimes heavy and difficult to follow due to many details regarding the study area. I have some doubts concerning the scientific contributions of such kind of studies, even though the publication could be justified by the importance of the study area and the

relevance of the river dynamics investigated. That said, the current manuscript fails in specifying what are the added knowledge provided compared to previous investigations performed in the same area, and with the same objective (see e.g. Tran et al., 2018). Hereafter some major and minor comments that need to be addressed before considering the manuscript for publication.

Major comments:

- One of the most relevant concern regarding the study is that it refers to only one year of data. The model has been calibrated and validated considering the 2000 and 2011 floods, respectively. After that, all the considerations regarding the river dynamics have been carried out referring to the event used for the calibration. However, Figure 5 clearly shows that the 2000 flood is different from the average condition. Thus, the question is: how representative is this event of the behaviors of the river network? The behavior of the different river branches and the way they interact depend on the specific contributions of the different basins. This to say that this analysis evaluates only a specific event, which might (is?) not be representative of the general river condition. As a matter of fact, previous studies investigating the same aspects (dyke effects) considered longer periods.

- The difference relative to the study of Tran et al. (2018) is sometimes cited in the document. However, the Authors should better specify the differences and the added knowledge ensured by this study. Also, are the results in line with previous findings? If not, how do you justify the difference? Does this study provide new information and knowledge relative to what was already known?

- Differences in terms of water elevation are in most of cases very minimal and of the same magnitude of the error of the model. What is the representativeness of such results. How can you exclude that those limited variations obtained among different configurations do not depend on the model itself, or on the way it reproduces the interaction between river and dyke rings?

- The overall model description should be improved. The model structure covers a key role in the overall study and additional details should be reported. The distinction between high and small dykes is not clear unless you are familiar with the study area. Please provide additional explanation about the model structure.

- Following all the hydrological details reported in the manuscript is sometimes difficult. Please always remind that a reader might not be familiar with the cited locations. All the cited locations should be identified within a Figure. For example, I would recommend adding a figure explaining the different geometric configurations and the dyke systems considered in the different cases. For example, what are the dyke system considered for the configuration LXQ, PoR, etc.?

Minor comments:

- P2L9: which kind of hydraulic structures?

- P3L15: please put the figure in relation to their citation order. Figure 4 is cited in the text before figures 2 and 3.

- P4L7-16: consider adding a scheme to better explain dykes interaction.

- Figure 2 and 3: why the number of dykes is significant? Probably reporting their overall length is more relevant.

- P7L24: I think here you should refer to "dike ring". How do you manage, within the model, areas partially protected with high dykes and partially not?

- Section 2.3: I would suggest using the term "configuration" instead of "scenario". Scenario is usually referred to identify different hydrological conditions (e.g. events of different return periods), while configuration sounds more appropriate for taking into account different topographic characteristics of the river network. Please make those locations clear using a map.

- Sometimes the structure of the paper is "heavy". Please consider to simplify it by

removing some sub-subsections, such as 2.4.1, 2.4.2. . .etc.

- P11L8: any discussion about the calibrated Manning coefficients: are they reasonable? Are they in agreement with those of previous studies?

- P13L31: Is it relevant reporting a difference of 0.6 cm? How reliable is this estimation? See also my previous comment on that.

- Figure 1: green areas should be reported as "high dyke protected areas". The same in Figure 4.

- Figure 7: check the unit of measure: m a.s.l. ?

- Figure 10-11: is the arrow dimension proportional to the flow? In case specify or add a legend.

- Figure 13: I was not able to find some of these stations in a map. Please add a reference to a map where those stations are visible.

---

## Author Comment (AC1) · 24 Apr 2019

**"Flooding in the Mekong Delta: Impact of dyke systems on downstream hydrodynamics" by Vo Quoc Thanh et al.**

**Responses to Referee #1's comments**

**Vo Quoc Thanh** (t.vo@un-ihe.org, vqthanh07@gmail.com)

Dear Referee #1,

Thank you so much for taking time to review and comment. We will consider your comments to revising the manuscript. The following section is our responses to your comments.

*The authors use a 1D/2D hydrodynamic model that covers the Mekong Delta including its rivers, major canals and extending into the continental shelf in the surrounding ocean, to investigate the impact of protecting agricultural areas with high-dykes on the river hydrodynamics. They found that (a) High dykes (particularly those in Long Xuyen Quadrangle (LXQ), Plains of Reeds (PoR)) have wide-spread impact on the flow downstream (b) has impact on inland tidal effects. Recent literature is well covered. Reasonably well-written introduction. Language use is generally adequate (though there are a number of technical issues that need correcting.) However, the paper is not easy to read as it is organized in such a way that a lot of (seemingly unnecessary) material is mixed with the main narrative of the paper.*

**Authors' response:** Thank you for the general comment on the paper. We will try to improve the paper structure to be easily readable.

*1. What is the benefit of modelling the continental shelf? This is not an oceanographic/coastal engineering study. Your focus (as stated) was to investigate the impact of construction of high-dykes on the flow regime of the river system. You are also not considering highly dynamic ocean impacts like storm surge. What is the drawback of stopping the model at the river mouth and providing tidal boundary conditions with sea level there? You might have good reasons for this approach. If so, they need to be explained.*

**Authors' response:** Regarding the modelling grid, we extended the modelling grid, including the shelf, in order to completely contain the river plume. Although the objective of this study is to investigate the impact of high-dyke constructions, we analyse hydrodynamics of the stations at the river mouths. In fact, the river discharge has contribution to sea levels (Kuang et al., 2017). Therefore, we included the shelf to investigate the impact and presented in the Table 4. If the modelling domain is limited at the river mouths, this impact would be excluded. As another reason, this approach is suitable toward salinity and sediment transport modelling.

*2. One year of simulation is a short period to obtain meaningful results. I think it is important to cover at least several years of flow data as such data for this case study is available. Is there a barrier to doing that?*

**Authors' response:** We partly agree with you that one year is a short period. However, it includes seasonal variation which is one of the main characteristics of the annual floods in the Mekong Delta. The model was used to compute for the floods in 2000 and 2001, but we analysed the flood in 2000. It is possible to run a multiple year simulation, but it is quite difficult to select a suitable period. We selected the flood in 2000 because it is one of the most severe floods recently. The highest water levels of the flood in 2000 is used as a reference for construction of flood prevention. Thus selecting a severe flood to evaluate impacts of high dykes could determine the maximum possible impacts of these construction on downstream hydrodynamics. To define a suitable period, it should consist of a low water flow year (2008), a moderate water flow year (2004) and high water flow year (2000) based on water levels at Tan Chau station. Another approach could use the long-term average hydrograph (MRC, 2009) as the boundary conditions.

*3. The point of departure (and justification for the methodology) of this paper seems to be the fact that previous studies could not able to predict the water level at the river mouth. If this is the sole justification to use a numerically expensive 2D model that includes continental shelf, the importance of obtaining those figures should be explained.*

**Authors' response:** As we responded to the comment, the major reason to include the shelf is to investigate the possible changes of water levels at the river mouths. The previous studies are usually 1D models and their boundaries are defined at the river mouths so they are not able to calculate water level changes at the river mouth stations.

*4. The point of doing a tidal harmonic analysis is unclear to me. Just testing the impact on the tidal range (amplitude) would have covered all the matter that is relevant to the central theme of the paper. Removing the tidal harmonic analysis part would shorten the paper - definitely would contribute to making it more readable and to the point.*

**Authors' response:** Thanks for the suggestion to make the paper more readable. However, results of the tidal harmonic analysis are important because hydrodynamics in the Mekong Delta downstream are strongly influenced by tides. Thus amplitudes of the eight main tidal constituents are suitable indexes to indicate tidal propagation changes. Using these indexes can eliminate effects of the fluvial floods.

*5. The authors should discuss the performance of the model. This is particularly important as many previous studies have used (much simpler) 1D modelling approach to arrive at similar results. How much is the computational effort? How does it compare with those reported in previous studies? What is the justification to use this modelling approach despite its expense (if that is the case)?*

**Authors' response:** We will revise the "Model performance" section. The modelling approach in this study overcome a limitation of the previous 1D models. This can help to understand hydrodynamics at the river mouths where rivers and oceans interact. We will compare the performance of this model with the previous models, but the model performance should reach a reasonable level. The model performance determined by the Nash-Sutcliffe efficiency (NSE). We compared the NSE values to the previous studies (e.g. Dang et al., 2018; Tran et al., 2017; Triet et al., 2017). NSE values of this model are slightly lower than the mentioned studies, but it is generally in the *good* category based

on (Moriasi et al., 2007) evaluation. The reason to use this modelling approach because its advantages, as mentioned in the comment 3's response.

*6. So many figures and many descriptions on model validation performance. This is an important topic to cover, but it is overdone in this case. Just one paragraph on how the model performed during validation and if absolutely necessary, one map showing validation results. Much of this can be moved to an appendix. In fact, it's best that they are presented as an online supplement rather than an Appendix, so as to keep the paper succinct and to the point.*

**Authors' response:** Thank you for your suggestion. We will take this comment to revise the paper.

*7. Scenarios need a better explanation. For example how much is protected with high dykes in "Dyke VMD" scenario? What is the basis?*

**Authors' response:** We will revise the paper with more details. The basic is the base scenario of the flood 2000, without high dykes. (Duong et al., 2016) found that there is no high dyke in the Vietnamese Mekong Delta before 2000.

*8. Water balance diagrams and descriptions are hard to understand. Please check the literature for much clearer ways of presenting these.*

**Authors' response:** Thank you for your suggestion.

*9. Lastly, it is important to place the findings within the context of other changes. Are these significant for example impact of climate change on upstream Mekong flow, dam construction, sea-level rise etc.? Some discussion on such issues is warranted.*

**Authors' response:** Thank you for comments.

**Comments on the manuscript**

Thanks for comments and suggestions on the manuscript. We will correct them.

**References**

Dang, D. T., Cochrane, T. A., Arias, M. E. and Dang, V. P.: Future hydrological alterations in the Mekong Delta under the impact of water resources development, land subsidence and sea level rise, J. Hydrol. Reg. Stud., 15(November 2017), 119–133, doi:10.1016/j.ejrh.2017.12.002, 2018.

Duong, V. H. T., Nestmann, F., Van, T. C., Hinz, S., Oberle, P. and Geiger, H.: Geographical impact of dyke measurement for land use on flood water in geographical impact of dyke measurement for land use on flood water in the Mekong Delta, in 8th Eastern European Young Water Professionals Conference - IWA, pp. 308–317., 2016.

Kuang, C., Chen, W., Gu, J., Su, T. C., Song, H., Ma, Y. and Dong, Z.: River discharge contribution to sea-level rise in the Yangtze River Estuary, China, Cont. Shelf Res., 134(June 2016), 63–75, doi:10.1016/j.csr.2017.01.004, 2017.

Moriasi, D. N., J. G. Arnold, M. W. Van Liew, R. L. Bingner, R. D. Harmel and T. L. Veith: Model Evaluation Guidelines for Systematic Quantification of Accuracy in Watershed Simulations, Trans. ASABE, 50(3), 885–900, doi:10.13031/2013.23153, 2007.

MRC: Annual Mekong Flood Report 2008, Vientiane., 2009.

Tran, D. D., van Halsema, G., Hellegers, P. J. G. J., Phi Hoang, L., Quang Tran, T., Kummu, M. and Ludwig, F.: Assessing impacts of dike construction on the flood dynamics in the Mekong Delta, Hydrol. Earth Syst. Sci. Discuss., 22, 1875–1896, doi:10.5194/hess-2017-141, 2017.

Triet, N. V. K., Dung, N. V., Fujii, H., Kummu, M., Merz, B. and Apel, H.: Has dyke development in the Vietnamese Mekong Delta shifted flood hazard downstream?, Hydrol. Earth Syst. Sci., 21(8), 3991–4010, doi:10.5194/hess-21-3991-2017, 2017.

---

## Author Comment (AC2) · 24 Apr 2019

**"Flooding in the Mekong Delta: Impact of dyke systems on downstream hydrodynamics" by Vo Quoc Thanh et al.**

**Responses to Referee #2's comments**

**Vo Quoc Thanh** (t.vo@un-ihe.org, vqthanh07@gmail.com)

Dear Referee #2,

Thank you so much for taking time to review and comment. We will consider your comments to revising the manuscript. The following section is our responses to your comments.

*Overview*

*The paper describes the effects of the high dykes structures on the complex interaction of the high flows and tides in Mekong Delta. The authors use 1D-2D model Delft3d Flexible Mesh model on unstructured grid to simulate the hydrodynamics at the river branches, canals and the adjacent ocean shelf with certain improvements compared to the previous studies done on Mekong Delta.*

*No major concerns about the language use, however the manuscript would benefit from the proofread by a native speaker. The research question is clearly stated and addressed in the Discussion section.*

**Authors' response:** Thank you so much for your suggestion.

*The main concerns*

*The overall structure of the manuscript is sufficient, however the abundance of details (some of those are unnecessary in my opinion) makes it difficult to read. The modelling part of the study needs major revisions (see comments below). Moreover, the figures should be revised. Therefore, I suggest major revisions of content further on.*

**Authors' response:** Thank you for your suggestion. We will revise them.

*General comments*

*Introduction. This section is somewhat well-structured and clear, however some information which is repetitive or not directly related to the study objective should be removed. For instance, speaking of the soil quality or challenges in agriculture may confuse the reader in the beginning. One sentence should be enough for it.*

**Authors' response:** Thank you for your suggestion. We will revise them.

*Methods. The following aspects need to be addressed:*

- *Input data. The authors use SRTM data to run the 2D part of the model. SRTM is known to have rather large vertical and horizontal errors, however, there is no overview related to the data pre-processing, resolution used and possible errors coming from the input data accuracy. Some of*

*the simulated water level changes are far smaller than the vertical accuracies of the terrain and bathymetry data. This aspect should be deliberately described in the Methods and Discussion section. The following paper might be useful: Hawker, L. P., Rougier, J., Neal, J. C., Bates, P. D., Archer, L., & Yamazaki, D. (2018). Implications of simulating global digital elevation models for flood inundation studies. Water Resources Research, 54.*
*https://doi.org/10.1029/2018WR023279.*

**Authors' response:** We only use the SRTM data for the floodplains. The bathymetry of 2D part for rivers were extract from the 1D ISIS model. The floodplains of the Mekong Delta are flat (Tran and Weger, 2017). The floodplain topography only influence during high flow season when the floodplains are inundated. Another reason is that the SRTM data for the Mekong Delta was efficiently used to simulate flood inundation (e.g. Dung Duc Tran et al. 2017; Triet et al. 2017). Therefore, we believe that the SRTM data are efficient to simulate the floods in the Mekong Delta.

- *High dyke geometry should be described (design return period, crest elevation, precise location etc.) and how they were incorporated into the computational mesh.*

**Authors' response:** The dyke geometry is various, as presented in the Figure 4 in the manuscript. The dyke geometries depend on the canal network. According to the Department of Agriculture and Rural Development of An Giang province, the high dykes are built, with their crest levels are higher than the peak water level of the flood 2000. Thus a high dyke in computation is defined as 2D dry grids. This modelling approach of floodplains and canals is based on field observations, as presented in Figure 1 and Figure 2.

[Figure]

Figure 1. The modelling grid. The floodplains are defined as 2D grids (in black) and canals are defined as 1D networks (in pink). The connecting links of 1D network and 2D grids are in blue.

[Figure]

Figure 2. A high dyke and non/low dyke in An Giang province and their schematization in modelling. Photos by Vo Quoc Thanh 2012.

- *Simulation time, if possible, should be increased to more years. One year is not sufficient enough to track the system behaviour in given setting and may cause biases.*

**Authors' response:** As the literature review, the major factor influences seasonal variation of fluvial flows is the Tonle Sap Lake storage. This feature is considered by using initial conditions which is simulated water levels of the previous flood.

- *The authors mention and illustrate throughout the manuscript high dyke as both, a flood protection measure and a dyke protected floodplain at the same time. This causes ambiguity and should be further specified/changed.*

**Authors' response:** We will correct it.

- *More details about the developed scenarios need to be added (see the dyke geometry).*

**Authors' response:** We will add more information about the developed scenarios.

*Results. The calibration/validation outcomes description should be substantially reduced, as it overloads the section. Computational time should be mentioned.*

**Authors' response:** Thank you. We will revise it.

*Discussion. I would recommend restructuring the discussion by answering the research questions in the same order as outlined in the Introduction section. It should be pointed out how the study benefited from using 2D code compared to 1D (MIKE11 or ISIS). The limitations related to data and methods used have to be put in a separate sub-section.*

**Authors' response:** Thank you for recommendations. We will revise it.

*Conclusion. Implications and future work should be added.*

**Authors' response:** Thank you for recommendations. We will revise it.

***Figures**. Geographical names and symbols should be consistent throughout the manuscript.*

**Authors' response:** Thank you for recommendations. We will correct them.

*P.2 line 17 – the sea level rise and land subsidence is an important point in the study area, the data about the future projections can be mentioned. However, it is important to point out why they are not considered in the paper (complexity, uncertainty, etc.).*

**Authors' response:** Thank you for comments. We will revise it.

*P.3 line 4 – sentence "These outcomes… " belongs to Conclusion*

*P.3 line 15- Figure 4 comes right after Figure 1*

*P.4 line 3 – the reference seems to be outdated*

*P.4 line 12 – return period of the event should be specified*

*P.4 line 14 – sentence "The water…" is ambiguous*

*P.4 line 31 – units should be specified*

**Authors' response:** Thank you for comments. We will correct them.

*P.7 line 21 - the grid density is mentioned to be sufficient, however the grid cell sizes seem to be quite large in fact. It would be useful to have some more explanation for the readers who are not familiar with Delft3DFM model.*

**Authors' response:** Thank you for comments. We will revise it.

*P.7 line 32 – floodplain bathymetry topography*

**Authors' response:** Thank you for comments. We will correct it.

*Table 1 can be removed*

**Authors' response:** Thank you for comments. We will correct it.

*Figure 1. The map is hard to read. I would recommend changing the following: reduce thickness of canal network, make the flood zones boundaries and fill more pronounced. The outline of municipalities (the faded orange line) should be removed. It is better to avoid overlap of green and red colours if possible*

**Authors' response:** Thank you for suggestion. We will edit the figure.

*Figure 2 and 3. What is meant here by high dykes? Dyke protected floodplains or flood defence? Should be clearly stated. Is there a way to combine two figures in one or incorporate in the Figure 4?*

**Authors' response:** We will clarify. The figure 2 and 3 present floodplain areas protected by high dykes until 2011.

*Figure 4. One of the north arrows should be removed. Region boundaries are not visible*

**Authors' response:** Thank you for comments. We will correct it.

*Figure 7. Everything that is above the sea-level (0m a.s.l.) is topography*

**Authors' response:** Thank you for comments. We will correct it.

*Figure 8 and 9. In legend the sign ">=" should be corrected*

**Authors' response:** Thank you for comments. We will correct it.

*Figure 10 and 11. The legend which specified the thickness of red arrows should be added*

**Authors' response:** Thank you for comments. We will correct it.

**References**

Tran, D. D. and Weger, J.: Barriers to implementing irrigation and drainage policies in An Giang Province, Mekong Delta, Vietnam, Irrig. Drain., doi:10.1002/ird.2172, 2017.

Tran, D. D., van Halsema, G., Hellegers, P. J. G. J., Phi Hoang, L., Quang Tran, T., Kummu, M. and Ludwig, F.: Assessing impacts of dike construction on the flood dynamics in the Mekong Delta, Hydrol. Earth Syst. Sci. Discuss., 22, 1875–1896, doi:10.5194/hess-2017-141, 2017.

Triet, N. V. K., Dung, N. V., Fujii, H., Kummu, M., Merz, B. and Apel, H.: Has dyke development in the Vietnamese Mekong Delta shifted flood hazard downstream?, Hydrol. Earth Syst. Sci., 21(8), 3991–4010, doi:10.5194/hess-21-3991-2017, 2017.

---

## Author Comment (AC3) · 24 Apr 2019

**"Flooding in the Mekong Delta: Impact of dyke systems on downstream hydrodynamics" by Vo Quoc Thanh et al.**

**Responses to Referee #3's comments**

**Vo Quoc Thanh** (t.vo@un-ihe.org, vqthanh07@gmail.com)

Dear Referee #3,

Thank you so much for taking time to review and comment. We will consider your comments to revising the manuscript. The following section is our responses to your comments.

*The study aims at evaluating the impact of high dykes on the hydrodynamic behavior of the Mekong river. The analysis is based on a 1D-2D model that reproduces the topographic characteristics of the river, as well as different dyke ring configurations. The work does not infer specific research questions but it is aimed at increasing the knowledge of the river dynamics and its behavior in the light of the recent construction of a complex dyke system. The manuscript is in general well written, even if it results sometimes heavy and difficult to follow due to many details regarding the study area. I have some doubts concerning the scientific contributions of such kind of studies, even though the publication could be justified by the importance of the study area and the relevance of the river dynamics investigated. That said, the current manuscript fails in specifying what are the added knowledge provided compared to previous investigations performed in the same area, and with the same objective (see e.g. Tran et al., 2018). Hereafter some major and minor comments that need to be addressed before considering the manuscript for publication.*

**Authors' response:** Thank you for your comment. We will respond to your comments and revise the manuscript.

*- One of the most relevant concern regarding the study is that it refers to only one year of data. The model has been calibrated and validated considering the 2000 and 2011 floods, respectively. After that, all the considerations regarding the river dynamics have been carried out referring to the event used for the calibration. However, Figure 5 clearly shows that the 2000 flood is different from the average condition. Thus, the question is: how representative is this event of the behaviors of the river network? The behavior of the different river branches and the way they interact depend on the specific contributions of the different basins. This to say that this analysis evaluates only a specific event, which might (is?) not be representative of the general river condition. As a matter of fact, previous studies investigating the same aspects (dyke effects) considered longer periods.*

**Authors' response:** Thank you for your comment. The model was used to compute for the floods in 2000 and 2001, but we analysed the flood in 2000. It is possible to run a multiple year simulation, but it is quite difficult to select a suitable period. We selected the flood in 2000 because it is one of the most severe floods recently. The highest water levels of the flood in 2000 are used as a reference for construction of flood prevention. Thus selecting a severe flood to evaluate impacts of

high dykes could determine the maximum possible impacts of these construction on downstream hydrodynamics.

*- The difference relative to the study of Tran et al. (2018) is sometimes cited in the document. However, the Authors should better specify the differences and the added knowledge ensured by this study. Also, are the results in line with previous findings? If not, how do you justify the difference? Does this study provide new information and knowledge relative to what was already known?*

**Authors' response:** This study provides a new modelling approach for the Mekong Delta which can overcome the limitation of the existing 1D models in order to simulate hydrodynamics at the river mouths. In addition, this approach may consider effects of coastal processes (e.g. waves, storm surge). Compared to the study of Tran et al. (2018), we included impacts of high dykes in PoR and TransBassac and the results of this study show a similar increase of water levels. Besides, we found that the dyked floodplains in the LXQ and PoR not only influence water regimes on its directly linked Mekong' branch, but also on the other branches. Moreover, we investigated the impacts of high dykes on tidal propagation along the Mekong River.

*- Differences in terms of water elevation are in most of cases very minimal and of the same magnitude of the error of the model. What is the representativeness of such results. How can you exclude that those limited variations obtained among different configurations do not depend on the model itself, or on the way it reproduces the interaction between river and dyke rings?*

**Authors' response:** We found that the water volume stored in the Vietnamese Mekong Delta's floodplains is much smaller than the annual flood volume. Thus this causes the small changes among the cases. To avoid the errors, the model spin-up time covers the 1999 flood. By using the same model setup, except the high dyke configuration, we believe that the differences are caused by high dykes.

*- The overall model description should be improved. The model structure covers a key role in the overall study and additional details should be reported. The distinction between high and small dykes is not clear unless you are familiar with the study area.*

*Please provide additional explanation about the model structure.*

**Authors' response:** Thank you for your comment. We will revise it. We only consider impacts of high dykes and assume that low dykes and non-dykes allow floodplain inundation. This modelling approach of floodplains and canals is based on field observations, as presented in Figure 1 and Figure 2.

[Figure]

Figure 1. The modelling grid. The floodplains are defined as 2D grids (in black) and canals are defined as 1D networks (in pink). The connecting links of 1D network and 2D grids are in blue.

[Figure]

Figure 2. A high dyke and non/low dyke in An Giang province and their schematization in modelling. Photos by Vo Quoc Thanh 2012.

*- Following all the hydrological details reported in the manuscript is sometimes difficult. Please always remind that a reader might not be familiar with the cited locations. All the cited locations should be identified within a Figure. For example, I would recommend adding a figure explaining the different geometric configurations and the dyke systems considered in the different cases. For example, what are the dyke system considered for the configuration LXQ, PoR, etc.?*

**Authors' response:** Thank you for your suggestion. We will check the mentioned locations and locate in a map. We will revise the Section 2.3 *High dyke development scenarios* with more details.

*- P2L9: which kind of hydraulic structures?*

**Authors' response:** We introduced the hydraulic structures in general. The hydraulic structures are commonly high dykes in the flood-prone areas and sluice gates for salinity prevention in the coastal areas.

*- P3L15: please put the figure in relation to their citation order. Figure 4 is cited in the text before figures 2 and 3.*

**Authors' response:** We will correct it.

*- P4L7-16: consider adding a scheme to better explain dykes interaction.*

**Authors' response:** We will add it.

*- Figure 2 and 3: why the number of dykes is significant? Probably reporting their overall length is more relevant.*

**Authors' response:** In my opinion, the high dykes are separated by the canal system. Thus the mean areas of floodplain protected by high dykes can reflex flood water conveyance. For example, in the same area of floodplain protected by high dykes, the smaller mean area has higher water conveyed capacity.

*- P7L24: I think here you should refer to "dike ring". How do you manage, within the model, areas partially protected with high dykes and partially not?*

**Authors' response:** In the case of a flood compartment containing a high dyke and a non-dyke, we used a ratio of areas of floodplains protected by the high dyke and the non-dyke. The larger area of floodplains will define the type of dykes.

*- Section 2.3: I would suggest using the term "configuration" instead of "scenario". Scenario is usually referred to identify different hydrological conditions (e.g. events of different return periods), while configuration sounds more appropriate for taking into account different topographic characteristics of the river network. Please make those locations clear using a map.*

**Authors' response:** Thank you so much for your suggestion. The study of (Tran et al., 2018) which was published in the HESS journal also use "scenario", so we prefer to use "scenario" to make it consistent.

*- Sometimes the structure of the paper is "heavy". Please consider to simplify it by removing some sub-subsections, such as 2.4.1, 2.4.2...etc.*

**Authors' response:** We will edit them.

*- P11L8: any discussion about the calibrated Manning coefficients: are they reasonable? Are they in agreement with those of previous studies?*

**Authors' response:** As mentioned in the calibration method section, we started to calibrate the model with calibrated roughness values from Manh et al. (2014) and Van et al. (2012). Thus the

calibrated roughness values are in agreement with these studies. However, there are slight differences in the coastal areas.

*- P13L31: Is it relevant reporting a difference of 0.6 cm? How reliable is this estimation?*

*See also my previous comment on that.*

**Authors' response:** This increase is reasonable because this is yearly mean increase.

*- Figure 1: green areas should be reported as "high dyke protected areas". The same in Figure 4.*

**Authors' response:** Thank you for your suggestion. We will edit it.

*- Figure 7: check the unit of measure: m a.s.l. ?*

**Authors' response:** Thank you for your suggestion. We will edit it.

*- Figure 10-11: is the arrow dimension proportional to the flow? In case specify or add a legend.*

**Authors' response:** We will add a legend.

*- Figure 13: I was not able to find some of these stations in a map. Please add a reference to a map where those stations are visible.*

**Authors' response:** Thank you for your suggestion. We will add it.

**References**

Manh, N. V., Dung, N. V., Hung, N. N., Merz, B. and Apel, H.: Large-scale quantification of suspended sediment transport and deposition in the Mekong Delta, Hydrol. Earth Syst. Sci. Discuss., 18, 3033–3053, doi:10.5194/hessd-11-4311-2014, 2014.

Tran, D. D., van Halsema, G., Hellegers, P. J. G. J., Phi Hoang, L., Quang Tran, T., Kummu, M. and Ludwig, F.: Assessing impacts of dike construction on the flood dynamics in the Mekong Delta, Hydrol. Earth Syst. Sci. Discuss., 22, 1875–1896, doi:10.5194/hess-2017-141, 2018.

Van, P. D. T., Popescu, I., Van Griensven, A., Solomatine, D. P., Trung, N. H. and Green, A.: A study of the climate change impacts on fluvial flood propagation in the Vietnamese Mekong Delta, Hydrol. Earth Syst. Sci., 16(12), 4637–4649, doi:10.5194/hess-16-4637-2012, 2012.

---

## Author Comment (AC4) · 18 Jun 2019

Dear Referee #1, Thank you so much for taking time to review and comment. We will consider your comments on the manuscript to revising the manuscript. The following section is our responses to your comments. Authors' response: The number of the annual flood volume at Kraie is estimated by about 416 km3. The reviewer#1 suggests the flood volume of 475 km3 is an estimate of the whole Mekong River. 1. what is the problem of dyke ring with mixed heights (obliviously the lowest point determines the level of protection) 2. Why it happens in this methodology (or what exactly happens: mixed height dyke rings misidentified as single height in the nodel?) 3. What is the relationship to ignoring small canals? Authors' response: The lowest elevation of a dyke ring determines the level. The canal system in the Vietnamese Mekong Delta

[Figure]

is dense, but the model considered the primary and secondary canals. The tertiary canals was excluded. Figure 1 presents an example of high dykes and low dykes. Unfortunately, the canal between low dykes and high dykes was excluded. The areas protected by high dyke is smaller than those protected by low dykes in the blue polygon (Figure 1b), so we assume that the blue polygon is defined as a low dyke in modelling.

Figure 1. An example of high dykes and low dykes; (a) high dykes in orange and low dykes in green; (b) the blue line presents a low dykes in modelling. (After Triet et al. 2017) is the kurtosis discussion necessary? This is useful if you do statistics of hundreds of hydrographs, but in this case we can clearly see that slipstream's are flatter by looking at the hydrographs. Authors' response: Although the different shapes of hydrographs are recognised by looking, the kurtosis helps to determine the differences. In this study's context, there is little value in doing a tidal harmonics analysis in my opinion. What you wanted to show is the high dyke development will affect the tidal impact on the river level in inland locations, while dykes do not affect it much in the coastal locations, as far as I could understand. A single graph showing the change in tidal range under different scenarios in various stations would show this adequately. Authors' response: As we responded to the comment 4, the results of tidal harmonic analysis help to understand tidal propagation which is indicated by the diurnal constituents (K1, O1, P1 and Q1) and the semidiurnal constituents (M2, K2, N2, and S2). In my opinion, using these indexes to present tidal variation is better than tidal ranges at which water levels are influenced by the annual floods. QLPH - is it a region or the project name? Please mark this area/infrastructure on a map (not at a new figure but in figure 4 or 1) and refer to it here. Authors' response: QLPH is a project which was constructed for water management in the coastal Mekong Delta (Hoanh et al. 2012). It is on the Figure 1.

[Figure]

**Fig. 1.**

---

## Author Comment (AC5) · 18 Jun 2019

Dear Editor and Reviewers,

Thank you so much for taking time to review our manuscript. We have responded to the reviewers' comments and revised the manuscript. We believe that the manuscript has been improved significantly. We are looking forward to hearing from you.

Sincerely, Vo Quoc Thanh

Please also note the supplement to this comment:
https://www.hydrol-earth-syst-sci-discuss.net/hess-2019-64/hess-2019-64-AC5-supplement.pdf

---

## Referee Report (RR1)

**Second revision of manuscript:** "*Flooding in the Mekong Delta: Impact of dyke systems on downstream hydrodynamics*".

**Overview and general remarks**

The second version of the manuscript has been improved compared to the first one. The authors put obvious effort in editing the text after the referee´s comments and suggestions. There are significant improvements in the Methods and Discussion/Conclusions section in terms of the content required for the study and readability. The Figures were improved. Nevertheless, the authors did not consider one of the most important remarks. The pre-processing of DEM and possible errors related to vertical accuracies of SRTM need to be described.

Large stripe noise and speckle noise have been extensively investigated in numerous studies (See Fig.1). For instance, Rodriquez et al 2006, Tarekegen et al 2013 and Kuenzer et al 2013 have reported that inaccuracies coming from data acquisitions for SRTM can cause significant errors in flood modelling, as such errors "… can seriously affect DEM derivatives such as slope, aspect and flow direction that affect the flow accumulation and consequently results in uncharacteristic channel network introducing a directional bias into subsequent flow routing in hydraulic and hydrologic modelling" (Tarekegen et al 2013). Recent advances of the Yamazaki et al 2017 have reported that original SRTM may have vertical errors to up to 10m.

One of the previous studies the authors referred to, Triet et al. 2017, used higher-resolution LiDAR DEM obtained from the Ministry of Natural Resources and Environment of Vietnam (MONRE), not SRTM.

First, in order to prove the validity of the results (sub-centimetre mean water level changes) the information on SRTM pre-processing and uncertainties coming from DEM data must be included in the manuscript. Without this crucial information, overall validity of the current study is compromised.

Second, I suggest authors to include a deliberate review on the previous studies done to model floods in Mekong delta (a table). Data used, domain modelled, modelling tools used.

Therefore, I suggest to return the manuscript for major revisions.

[Figure]

Figure 1. Image representing original SRTM data over Mekong data. Taken from http://hydro.iis.u-tokyo.ac.jp/~yamadai/MERIT_DEM/.

**References**

Kuenzer, C., Guo, H., Huth, J., Leinenkugel, P., Li, X., & Dech, S. (2013). Flood mapping and flood dynamics of the Mekong Delta: ENVISAT-ASAR-WSM based time series analyses. *Remote Sensing*, *5*(2), 687-715.

Rodriguez, E., Morris, C. S., & Belz, J. E. (2006). A global assessment of the SRTM performance. *Photogrammetric Engineering & Remote Sensing*, *72*(3), 249-260.

Tarekegn, T. H., & Sayama, T. (2013). Correction of SRTM DEM artefacts by Fourier transform for flood inundation modeling. *Journal of Japan Society of Civil Engineers, Ser. B1 (Hydraulic Engineering)*, *69*(4), I_193-I_198.

Triet, N. V. K., Nguyen, V. D., Fujii, H., Kummu, M., Merz, B., & Apel, H. (2017). Has dyke development in the Vietnamese Mekong Delta shifted flood hazard downstream?. *Hydrology and Earth System Sciences*, *21*(8), 3991.

Yamazaki, D., Ikeshima, D., Tawatari, R., Yamaguchi, T., O'Loughlin, F., Neal, J. C., ... & Bates, P. D. (2017). A high-accuracy map of global terrain elevations. *Geophysical Research Letters*, *44*(11), 5844-5853.

---

## Author Response (AR2)

[revised manuscript text omitted]

**Responses to Referee #2's comments**

Overview and general remarks

The second version of the manuscript has been improved compared to the first one. The authors put obvious effort in editing the text after the referee´s comments and suggestions. There are significant improvements in the Methods and Discussion/Conclusions section in terms of the content required for the study and readability. The Figures were improved. Nevertheless, the authors did not consider one of the most important remarks. The pre-processing of DEM and possible errors related to vertical accuracies of SRTM need to be described.

**Author response**: Thank you for your review. The remaining comments would be answered below.

Large stripe noise and speckle noise have been extensively investigated in numerous studies (See Fig.1). For instance, Rodriquez et al 2006, Tarekegen et al 2013 and Kuenzer et al 2013 have reported that inaccuracies coming from data acquisitions for SRTM can cause significant errors in flood modelling, as such errors "… can seriously affect DEM derivatives such as slope, aspect and flow direction that affect the flow accumulation and consequently results in uncharacteristic channel network introducing a directional bias into subsequent flow routing in hydraulic and hydrologic modelling" (Tarekegen et al 2013). Recent advances of the Yamazaki et al 2017 have reported that original SRTM may have vertical errors to up to 10m.

One of the previous studies the authors referred to, Triet et al. 2017, used higher-resolution LiDAR DEM obtained from the Ministry of Natural Resources and Environment of Vietnam (MONRE), not SRTM.

First, in order to prove the validity of the results (sub-centimetre mean water level changes) the information on SRTM pre-processing and uncertainties coming from DEM data must be included in the manuscript. Without this crucial information, overall validity of the current study is compromised.

**Author response**: As we observed, the floodplains in the Vietnamese Mekong Delta are quite flat (Figure 1). The elevation of these floodplains are from 0-4 m (Tuan et al. 2007). In fact, the flood

peak at Tan Chau station was about 5.2 m in 2000 and it causes inundation of these floodplains. Therefore, samples from SRTM DEM which have higher 5.2 m elevation were eliminated, were used for the VMD's floodplain. In our modelling approach, the floodplains in the VMD is represented by large 2D cells and small canals in these floodplains are excluded. Thus these floodplains mainly have a function of water storage during the high flow season. We do not use the SRTM DEM to generate the channel network, but the channel network and river topography were extracted from the 1D-ISIS model. I totally agree with the reviewer comment on the errors coming from the SRTM DEM and they should be assessed, but this needs ground truth data. Thus the possible errors of the SRTM DEM was reviewed and this limitation is also highlighted.

[Figure]

Figure 1. A floodplain in Chau Phu district, An Giang province (Vo Quoc Thanh, 2012).

Second, I suggest authors to include a deliberate review on the previous studies done to model floods in Mekong delta (a table). Data used, domain modelled, modelling tools used.

**Author response**: The previous studies on modelling floods in the Mekong Delta were reviewed and discussed in the manuscript.

Therefore, I suggest to return the manuscript for major revisions.

[Figure]

Figure 1. Image representing original SRTM data over Mekong data. Taken from http://hydro.iis.u-tokyo.ac.jp/~yamadai/MERIT_DEM/.

**Author response**: I hope that our answers and responses address your comments.

**Responses to Referee #3's comments**

The Authors have improved the manuscript, which I believe is now almost ready for publication. However, they have partially faced some of my previous comments. I believe some clarifications are needed:

**Author response**: Thank you very much for your comments. These comments were responded below.

- Accuracy of the model: I think the Authors have misunderstood my previous comment (at least looking at their reply), which is reported below:

"Differences in terms of water elevation are in most of cases very minimal and of the same magnitude of the error of the model. What is the representativeness of such results? How can you exclude that those limited variations obtained among different configurations do not depend on the

model itself, or on the way it reproduces the interaction between river and dyke rings?"

**Author response**: I clearly understand your comment. This is an interesting question. The fact that the differences in the cases are minimal and relatively equal to the model error. In order to separate the model error due to model setup of dyke rings and rivers, uncertainties of dyke rings need to be quantified. This limitation is added in the manuscript.

Errors of the model after calibration are comparable to the variability of water levels founded among different scenarios, if not even larger at some locations. Let's say, if your model is on average precise with an error of +- 15 cm, what is the reliability of differences of 0.0x cm (see Table 2). I know it is difficult to establish the accuracy of your model but, at least, this point must be clearly pointed out. Your model can be used in relative terms, while you should be careful when providing specific water levels and discharges.

**Author response**: This is a great suggestion. We would like to include it in the manuscript. We did compare the model errors and the differences among scenarios at some selected stations. Due to the fact that the differences of water levels among the scenarios are small, the differences are fallen into the model error variations, except at Tanchau station (Figure 2).

[Figure]

Figure 2. A statistical comparison of model errors and differences among the scenarios.

This comment is also related to another one risen by a different Reviewer, who highlight the uncertainty of topographic data (SRTM) adopted for the model. Although SRTM is used only for the floodplains, the errors on this data are not negligible and they can significantly affect the simulation results. This limitation should also be highlighted, emphasizing the fact that the model enables to better understand the dynamics of the system and might serve as a tool for comparative studies.

**Author response**: The limitation of SRTM data would be included in the manuscript.

Finally, I haven't understood the reply to my following comment; can you please further elaborate the concept?

- Comment: "Figure 2 and 3: why the number of dykes is significant? Probably reporting their overall length is more relevant"

- Authors' response: In my opinion, the high dykes are separated by the canal system. Thus the mean

areas of floodplain protected by high dykes can reflex flood water conveyance. For example, in the same area of floodplain protected by high dykes, the smaller mean area has higher water conveyed capacity.

**Author response**: I think the overall length and the mean area of floodplain protected by high dykes are similar to indicate the channel density in the floodplain in the case of the floodplain completely protected by high dykes. However, we introduced the case of mixed high and low dykes in the Vietnamese Mekong Delta, so calculating the total length of dyke rings is difficult. Therefore, we used the indicator of the mean area of floodplain protected by high dykes.

---

## Author Response (AR3)

**"Flooding in the Mekong Delta: Impact of dyke systems on downstream hydrodynamics" by Vo Quoc Thanh et al.**

**Vo Quoc Thanh** (t.vo@un-ihe.org, vqthanh07@gmail.com)

Dear Editor,

Thank you so much for suggestion to improve the manuscript. We have revised the manuscript by adding uncertainty discussion. We would like to thank the editor and three anonymous reviewers for leaving a review of our manuscript. We believe that the scientific quality of this manuscript was significantly improved. We hope you find it suitable for publication and are looking forward to hearing from you.

Sincerely,

Thanh